# Characterisation of the OTU domain deubiquitinase complement of *Toxoplasma gondii*

Mary-Louise Wilde[1], Ushma Ruparel[1], Theresa Klemm[1], V Vern Lee[1,2], Dale J Calleja[1], David Komander[1], Christopher J Tonkin[1]

**The phylum Apicomplexa contains several parasitic species of medical and agricultural importance. The ubiquitination machinery remains, for the most part, uncharacterised in apicomplexan parasites, despite the important roles that it plays in eukaryotic biology. Bioinformatic analysis of the ubiquitination machinery in apicomplexan parasites revealed an expanded ovarian tumour domain–containing (OTU) deubiquitinase (DUB) family in *Toxoplasma*, potentially reflecting functional importance in apicomplexan parasites. This study presents comprehensive characterisation of *Toxoplasma* OTU DUBs. AlphaFold-guided structural analysis not only confirmed functional orthologues found across eukaryotes, but also identified apicomplexan-specific enzymes, subsequently enabling discovery of a cryptic OTU DUB in *Plasmodium* species. Comprehensive biochemical characterisation of 11 *Toxoplasma* OTU DUBs revealed activity against ubiquitin- and NEDD8-based substrates and revealed ubiquitin linkage preferences for Lys6-, Lys11-, Lys48-, and Lys63-linked chain types. We show that accessory domains in *Toxoplasma* OTU DUBs impose linkage preferences, and in case of apicomplexan-specific TgOTU9, we discover a cryptic ubiquitin-binding domain that is essential for TgOTU9 activity. Using the auxin-inducible degron (AID) to generate knockdown parasite lines, TgOTUD6B was found to be important for *Toxoplasma* growth.**

## Introduction

Ubiquitination is a highly conserved post-translational modification found across archaea, bacteria, and eukaryotes and plays important roles in nearly every cellular process. Protein modification with ubiquitin takes on a multitude of topological states and can be thought of as a code—written and read by the cell—within which the fates of protein substrates are mapped out (Swatek & Komander, 2016). Ubiquitin is a small 76–amino acid protein commonly attached to lysine residues of protein substrates in an enzymatic cascade. Briefly, ubiquitin is activated by an E1 ubiquitin–activating enzyme and transferred to an E2 ubiquitin–conjugating enzyme. An E3 ubiquitin ligase mediates the addition of ubiquitin to a substrate protein. Once the first ubiquitin molecule is attached, it can be further ubiquitinated at one of its seven lysine residues or at the N-terminal methionine, generating polyubiquitin chains. Differentially linked ubiquitin moieties create chains with distinct architectures that determine downstream signalling outcomes for the modified protein. For example, Lys48-linked ubiquitin chains target a protein for degradation, whereas Lys63 ubiquitination plays important roles in nuclear factor kappa-light-chain-enhancer of activated B cell (NFκB) signalling (Tracz & Bialek, 2021). Ubiquitin is antagonised by deubiquitinases (DUBs), ubiquitin-specific proteases that are responsible for removing or trimming ubiquitin chains from substrates to create a highly dynamic system. DUBs maintain cellular ubiquitin levels by processing newly synthesised ubiquitin precursors and enable recycling of ubiquitin from proteins targeted for proteasomal degradation. DUBs also play key roles throughout the cell, regulating protein homeostasis directly and indirectly via maintenance of ubiquitination states. The association of deregulated DUB activity with several disease states including cancers, immunopathologies, and neurodegenerative diseases has led to numerous drug discovery programmes targeting DUBs (Clague et al, 2019).

The phylum Apicomplexa represents a group of divergent parasitic protists comprising multiple species of medical and agricultural importance: *Plasmodium* parasites cause malaria and are responsible for over half a million deaths each year in mostly low-income countries (World Health Organization, 2021); *Cryptosporidium* parasites cause cryptosporidiosis—a leading cause of childhood diarrhoea and malnutrition worldwide (Pane & Putignani, 2022); and *Toxoplasma gondii*, which causes toxoplasmosis and establishes a latent infection, which can be life-threatening in immunocompromised patients (Montoya & Liesenfeld, 2004). These eukaryotic intracellular parasites encode their own ubiquitination machinery; however, the role of ubiquitination in parasite homeostasis and pathogenesis is still unclear. Simple eukaryotes such as protists may be beneficial to derive novel and fundamental features of the

[1]Walter and Eliza Hall Institute of Medical Research, Parkville, Australia; and Department of Medical Biology, University of Melbourne, Melbourne, Australia   [2]Bio21 Molecular Science and Biotechnology Institute, Parkville, Australia; and Department of Biochemistry and Pharmacology, The University of Melbourne, Melbourne, Australia

Correspondence: dk@wehi.edu.au; tonkin@wehi.edu.au

ubiquitin system in more divergent and specialised organisms. Along with its high genetic tractability, *Toxoplasma* comprises several desirable members that make it the model apicomplexan of choice to study features that may be conserved across the phylum (Dubey, 2020).

The *Toxoplasma* genome encodes all the necessary components for the ubiquitin–proteasome system including one E1 enzyme, 13–16 predicted E2-conjugating enzymes and ~72 E3 ligases, and DUBs and other scaffolding machinery (Ponts et al, 2008). Furthermore, *Toxoplasma* encodes the necessary components for protein modification with other ubiquitin-like proteins (UBLs), including ubiquitin-related modifier 1 (Urm1), neuronal precursor cell–expressed developmentally down-regulated protein 8 (NEDD8), autophagy-related protein 8 (Atg8), and small ubiquitin-like modifier (SUMO), and these pathways appear to be functional (Frickel et al, 2007; Kong-Hap et al, 2013; Crater et al, 2018; Tan et al, 2022). Studies investigating the apicoplast—a secondary plastid of endosymbiotic origin—have revealed further interesting adaptations in apicomplexan ubiquitination machinery. Apicoplast-localised ubiquitination enzymes derived from endoplasmic reticulum–associated degradation components, and an apicoplast-specific ubiquitin-like protein, PUBL, were discovered to be essential for the import of nuclear-encoded proteins into the *Toxoplasma* apicoplast (Agrawal et al, 2013; Fellows et al, 2017).

Few apicomplexan DUBs have been characterised, and a global understanding of DUB biology in *Toxoplasma* is lacking. DUBs can be classified into seven distinct families, six of which are cysteine proteases: the ubiquitin-specific proteases (USP/UBP); ubiquitin C-terminal hydrolases (UCH); ovarian tumour proteases (OTU); Josephins; and the recently discovered MINDY and zinc finger with UFM1-specific peptidase domain protein (ZUP1) families (Clague et al, 2019). A seventh family comprises the JAB1/MPN/MOV34s (JAMM) metalloproteases. The USP, UCH, OTU, Josephin, MINDY, and JAMM families are conserved in *Toxoplasma*; however, few have been characterised (Ponts et al, 2008).

OTU deubiquitinases are of particular interest as they often display diversity in their linkage specificity (Mevissen et al, 2013; Schubert et al, 2020) and have non-degradative roles in cell signalling processes (Du et al, 2019). Five apicomplexan linkage-specific OTU DUBs have been investigated and have been implicated in non-degradative cellular processes including apicoplast homeostasis and parasite development (Ju et al, 2014; Dhara & Sinai, 2016; Datta et al, 2017; Wang et al, 2017, 2018, 2019). *Tg*OTUD3A is the only *Toxoplasma* OTU DUB characterised to date and was found to preferentially cleave Lys48-linked polyubiquitin chains when tested against a subset of chain types (Dhara & Sinai, 2016). Comprehensive characterisation of the OTU DUBs found in apicomplexans is, however, lacking. Understanding the linkage specificity of the OTU complement in the context of *Toxoplasma* would provide an opportunity for a better understanding of their biological roles and may give insights into the broader functions of OTU DUBs across the phylum.

In this study, we have defined the OTU deubiquitinase family in *Toxoplasma*. Through comprehensive biochemical characterisation, we reveal linkage specificity profiles for members of this family. Strikingly, these are divergent from other studied eukaryotes, with many members showing a profile resembling bacterial

OTU DUBs. AlphaFold structural prediction (Jumper et al, 2021; Mirdita et al, 2021 *Preprint*) has enabled analysis of the *Toxoplasma* OTU domain architecture and identified unique features in catalytic and accessory domains of *Toxoplasma* OTU DUBs, including a cryptic ubiquitin-binding domain (UBD) essential for TgOTU9 specificity and function. Furthermore, through auxin-inducible degron (AID)–mediated knockdown we reveal TgOTUD6B to be required for parasite survival in vitro. Together, our results demystify this family of DUBs in *Toxoplasma*, showing that OTU DUBs are important for *Toxoplasma* biology and virulence and may serve as future drug targets.

## Results

### Annotation of the OTU complement in *T. gondii*

It is well appreciated that the number of DUBs in distinct families varies greatly across organisms. Compared with one to five members in *Plasmodium*, *Cryptosporidium*, and other simple eukaryotes, *Toxoplasma* was previously found to encode 10–12 OTU domain–containing genes (Ponts et al, 2008; Dhara & Sinai, 2016). Combining prior studies and further sequence annotation, we identified 14 OTU DUBs in *Toxoplasma*; a main difference to previous studies was the reannotation of OTUD1 orthologues (Fig 1). Furthermore, this trend of OTU DUB expansion can be extended to other cyst-forming coccidian genomes (Fig 1A).

OTU DUBs can be subclassified into four subfamilies: OTUBs/ Otubains, OTUDs, A20-like OTU DUBs, and the OTULIN subfamily (Mevissen et al, 2013). *Toxoplasma* comprises one OTUB1 enzyme, whereas the remaining OTU DUBs are in the OTUD subfamily. With the exception of TgOTUD3A and TgOTUD3C, their minimal catalytic domains comprise insertions of varying length and at distinct points (Fig 1B).

Long disordered regions are also present outside of the OTU domain in most members, which are commonly seen in *Toxoplasma* and other early-branching eukaryotes (Mohan et al, 2008). Interestingly, three members are predicted to contain a transmembrane domain, a feature not seen in human OTU DUBs (Mevissen et al, 2013). A further distinguishing feature is the relative lack of accessory domains, including UBDs, with two notable exceptions. TgOTUD2 shows the conserved domain architecture seen in orthologous OTUD2 proteins, with an N-terminal UBX-like domain and a C-terminal ZnF-C2H2 domain (Fig 1B). We also discovered that TgOTU9, which appears to have orthologues only in apicomplexans and in some pathogenic fungi, comprises a cryptic yet functional UBD. The earliest branching OTU DUB in *Toxoplasma*, TGME49_268690, was named TgOTU10. Phylogenetic analysis shows that orthologues of this enzyme are only found in other apicomplexans (Table 1).

We then assessed the functional importance of OTU DUBs using the recent *Toxoplasma* genome-wide CRISPR screens. Using the *Toxoplasma* online genome resource ToxoDB (Harb & Roos, 2020), we found five of 14 OTU DUBs show highly negative CRISPR "fitness scores" (range: −2.5 to −4.4), indicating functional importance (Table 1) (Sidik et al, 2016). Data from a study employing hyperplexed localisation of organelle proteins by isotope tagging (hyperLOPIT) were

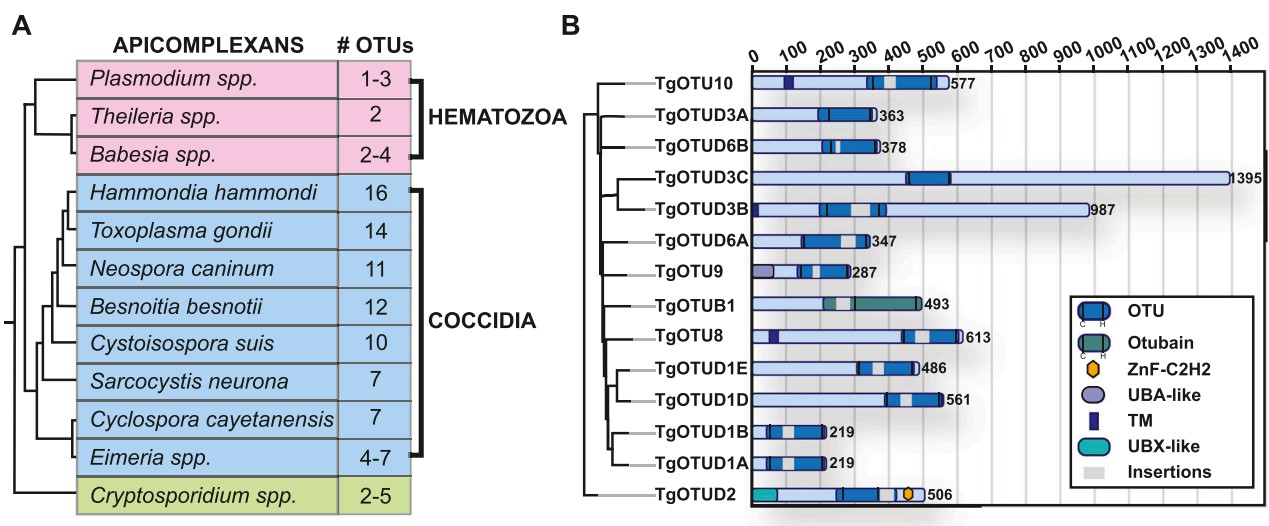

**Figure 1. OTU complement in *Toxoplasma gondii*.**
**(A)** Numbers of OTU-encoding genes in apicomplexans. Phylogeny derived from Mathur et al (2021). **(B)** Domain composition of *Toxoplasma* OTU DUBs. Domain-encoding regions were identified with PROSITE, and multiple sequence alignments with related orthologues were used to determine domain boundaries and insertions (Sigrist et al, 2013). Grey bars indicate disordered insertions identified with IUPred2A (Mészáros et al, 2018; Erdős & Dosztányi, 2020).

available for six of 14 genes and revealed a range of localisation including the apicoplast (Barylyuk et al, 2020) (Table 1). Together, the expanded *Toxoplasma* OTU complement appears to comprise a range of functionally important enzymes, some with predictable but several with unclear roles within this simple eukaryote.

## AlphaFold-based annotation of OTU DUBs in *Toxoplasma*

With the recent revolutionary deep-learning algorithms provided by AlphaFold, protein structures can be predicted by sequence with high confidence (Jumper et al, 2021; Mirdita et al, 2021 *Preprint*). We predicted structures for full-length *Toxoplasma* OTU DUBs (Fig 2).

**Table 1. Publicly available data compiled from ToxoDB.**

| Gene ID | Product description | CRISPR phenotype score (Sidik et al, 2016)[a] | Predicted localisation (Barylyuk et al, 2020)[b] | Previous name (Dhara & Sinai, 2016) | Updated name[c] |
|---|---|---|---|---|---|
| TGME49_207650 | OTU family cysteine protease | −2.75 | N/A | TgOTUD1A | TgOTUD1A |
| TGME49_237894 | OTU family cysteine protease | −4.37 | N/A | TgOTUD1B | TgOTUD1B |
| TGME49_210678 | OTU family cysteine protease | N/A | N/A | N/A | TgOTUD1D |
| TGME49_237900 | OTU family cysteine protease | N/A | N/A | N/A | TgOTUD1E |
| TGME49_277990 | OTU family cysteine protease | −0.13 | Nucleus—chromatin | TgOTUD2 | TgOTUD2 |
| TGME49_258780 | OTU family cysteine protease | 0.74 | PM—peripheral 2 | TgOTUD3A | TgOTUD3A |
| TGME49_229710 | OTU family cysteine protease | −3.46 | N/A | TgOTUD3B | TgOTUD3B |
| TGME49_243430 | OTU family cysteine protease | 0.43 | PM—peripheral 2 | TgOTUD3C | TgOTUD3C |
| TGME49_271070 | Hypothetical protein | −0.23 | N/A | TgOTU7 | TgOTUD6A[d] |
| TGME49_243510 | OTU family cysteine protease | −2.99 | Cytosol, 19S proteasome | TgOTUD5 | TgOTUD6B[d] |
| TGME49_260510 | Ubiquitin thioesterase otubain-like family protein | 1.48 | Cytosol | TgOTUB1 | TgOTUB1 |
| TGME49_216440 | OTU family cysteine protease | 0.38 | N/A | TgOTU8 | TgOTU8 |
| TGME49_266500 | Hypothetical protein | 0.89 | N/A | TgOTU9 | TgOTU9 |
| TGME49_268690 | Hypothetical protein | −2.47 | Apicoplast | N/A | TgOTU10 |

[a]CRISPR phenotype score indicates functional importance of genes after using a genome-wide knockout screen (more negative = more fitness conferring) (Sidik et al, 2016).
[b]Localisation data from hyperplexed localisation of organelle proteins by isotope tagging (hyperLOPIT) (Barylyuk et al, 2020).
[c]Updated naming for this study (see the Materials and Methods section).
[d]Renamed OTU family members respective to Dhara and Sinai (2016).

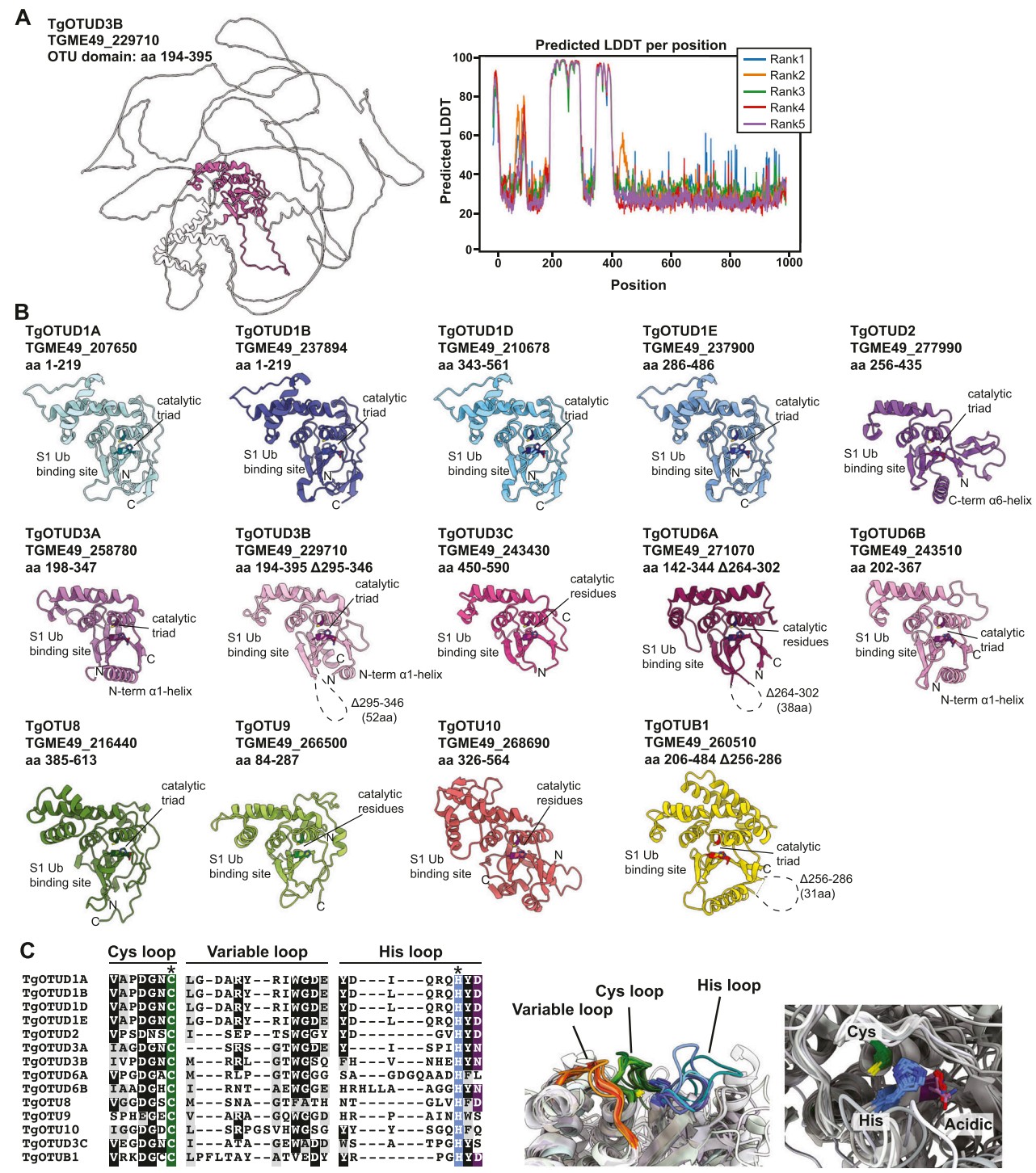

**Figure 2. Structural prediction of *Toxoplasma* OTU DUBs using AlphaFold.**
**(A)** Left panel: AlphaFold structural prediction of full-length TgOTUD3B. OTU domain is highlighted in pink. Right: predicted LDDT indicates confidence of a prediction at a given residue. Regions of low confidence (pLDDT < 50) are represented as unstructured loops (Source: ColabFold AlphaFold2_advanced notebook [Jumper et al, 2021; Mirdita et al, 2021 *Preprint*]). **(B)** AlphaFold predictions of *Toxoplasma* OTU domains highlighting S1 Ub-binding sites, catalytic triad placement, N- and C-termini (denoted by N and C, respectively), and structurally equivalent N- or C-terminal α-helices, if present. Dashed lines represent disordered regions of low confidence (pLDDT < 50). **(C)** Left panel: sequence of Cys, His, and variable loops in *Toxoplasma* OTU enzymes. Asterisks (*): catalytic Cys and His residues. Right panel: superposition of Cys, His, and variable loops (left), and catalytic triad residues (right).

Predicted local-distance difference test (pLDDT) values indicate the accuracy of a structural prediction at a given residue; OTU domains were predicted with high confidence (pLDDT > 70), whereas disordered regions both within and around the OTU domains (pLDDT < 50) could not be confidently predicted and are likely unstructured elements (Fig 2A) (Jumper et al, 2021; Mirdita et al, 2021 Preprint). As expected, all sequence-derived TgOTU DUBs (Fig 1B) contain an easily recognisable OTU fold (Fig 2B). Superposition of TgOTU structural predictions shows a competent placement of catalytic triad Cys-His-Asp/Asn residues (Fig 2C). TgOTUD3C, TgOTUD6A, TgOTU9, and TgOTUD10 lack an Asp/Asn residue +2 upstream of the catalytic His; however, this residue is not always essential for OTU activity (Komander & Barford, 2008; Schubert et al, 2020) (Fig 2C).

*Toxoplasma* OTU domains are variable in the S1 distal Ub-binding site, comprising extended or additional α-helices (TgOTUD1 orthologues, TgOTU8, TgOTU9, and TgOTU10), or small insertions that shape the S1 site (TgOTUD3A and TgOTUD6A) (Figs 2B and S1A–E). Long disordered insertions are also seen in the central β-sheet of TgOTUD6A and TgOTUD3B of 38 and 52 residues, respectively, and TgOTUB1 has a 31-residue disordered region inserted after the N-terminal α-helix (Figs 2B and S1E–G). Protein surface conservation analysis based on alignments in Supplemental Data 1. revealed strong conservation of residues in the S1' Ub-binding site (Fig S2) (Ashkenazy et al, 2016). The S1' site comprises several key structural elements: the catalytic Cys and His loops, a third variable loop, and N- or C-terminal α-helices that make contacts with the proximal ubiquitin molecule (Mevissen et al, 2013). Alignment of these loops reveals conservation particularly in the Cys and variable loops of TgOTU DUBs, whereas the His loop varies in length and is extended in TgOTUD6A and TgOTUD6B (Fig 2C). The N-terminal α1-helix is only present in TgOTUD3A, TgOTUD3B, and TgOTUD6B. TgOTUD2 has a structurally equivalent C-terminal α6-helix (Fig 2B).

### Identification of structural orthologues of *Toxoplasma* OTU DUBs

The high accuracy of protein structure prediction by AlphaFold provided us the opportunity of annotating the OTU DUBs according to their conserved structural elements, in addition to the sequence. After these new annotations, we sought to use the wealth of structural information deposited into the AlphaFold Protein Structure Database (Varadi et al, 2021) to discover new OTU domain–containing DUBs. First, we set out to identify cross-species similarities for *Toxoplasma* OTU domains (Table 2). As mentioned above, TgOTUD2 and TgOTUB1 are highly conserved members with clear similarities to human counterparts. In contrast, no human orthologues are readily identifiable for TgOTU8, TgOTU9, and TgOTU10 via sequence-based BLAST searches.

To identify structural orthologues, the DALI protein structure comparison server was used to compare AlphaFold predicted OTU structures against the Protein Data Bank (PDB) (Holm, 2022). Multiple OTU domains have been crystallised including human OTUD1 and OTUD3, which comprise a minimalistic, highly conserved catalytic core (Mevissen et al, 2013). Both align well with all *Toxoplasma* OTU models (Fig S1A, B, and D–F), and DALI confirms human OTUD3 (PDB ID: 4BOU) as the top structural hit for most *Toxoplasma* OTU domains, including for TgOTU9 (Z-scores 12.8–19.4) (Table 2). TgOTU10 best aligned with bacterial wMelOTU from *Wolbachia pipientis* (PDB ID: 6W9R; Z-score 13.0) (Table 2 and Fig S1C) (Schubert et al, 2020).

Next, we performed a similar DALI search against the human AlphaFold database (Varadi et al, 2021; Holm, 2022). Top hits for most TgOTU DUBs from the human AlphaFold database were OTUD1 (Z-scores 12.2–19.6) (Table 2).

TgOTUD6A aligned best with human OTUD6B (Z-score 13.2), and TgOTUD6B aligned well with both human OTUD6A and OTUD6B (Z-scores 22.1 and 21.2). OTUD6B is among the highest conserved OTU DUBs in eukaryotes; however, its function is unclear, and despite a predicted functional active site, it tends to display no or very low activity against polyubiquitin substrates. Similarly, the top hits for TgOTUD3C and TgOTUD3B were OTUD5 and ALG13, respectively (Z-scores 16.8, 17.8) (Table 2). Both enzymes are inactive in vitro, and OTUD5 requires an activating phosphorylation event (Huang et al, 2012; Mevissen et al, 2013).

### Discovery of cryptic DUBs in other parasites

DALI also enables comparison against a subset of other eukaryotic and bacterial AlphaFold databases (Table 2) (Holm, 2022), which was particularly interesting for the evolutionarily distant members such as TgOTU10. A DALI search of TgOTU10 against proteins of *Cryptosporidium parvum* identified an orthologous protein, cgd7_200, that lacks domain annotations. AlphaFold structural prediction of cgd7_200 reveals a similar fold to TgOTU10 (RMSD 0.975) (Fig S1H). More strikingly, AlphaFold predicted a split OTU domain protein in *Plasmodium falciparum*. A DALI search of TgOTU10 against predicted *P. falciparum* proteins unveiled high similarities (Z-score 20.0) with PF3D7_1350500, an uncharacterised protein that lacks domain annotation and did not come up in sequence-based homology searches. In PF3D7_1350500, residues 282–428 interact with C-terminal residues 1,026–1,144 to make an OTU fold that aligns reasonably well with TgOTU10 (RMSD 1.187) (Fig S1I and J). This discovery once again highlights the power and versatility of AlphaFold to discover functional domains, extending to split enzymes unrecognised in genomes (Jumper et al, 2021; Varadi et al, 2021). Together, the combination of sequence and structural alignments is likely to give a superior functional framework for the annotation of enzyme classes in less studied organisms.

### Activity of *Toxoplasma* OTU DUBs against ubiquitin

The definition of the OTU DUB complement of *Toxoplasma* was followed with comprehensive characterisation of OTU activities. Full-length constructs of TgOTUD1D, TgOTUD2, TgOTUD3A, TgOTUD6A, TgOTUD6B, and TgOTU9, and OTU domain–containing constructs of TgOTUD3B, TgOTUD3C, TgOTUB1, TgOTU8, and TgOTU10 were expressed and purified from *Escherichia coli* for biochemical characterisation (Fig 3A). Fig S3A shows additional full-length constructs that despite best efforts could not be expressed. As a first test for reactivity with ubiquitin, we used Ub–propargylamine (Ub-PA), an activity-based probe that covalently reacts with the catalytic cysteine of a DUB (Ekkebus et al, 2013). The resulting 8.5-kD

**Table 2.** Sequence and structural orthologues of *Toxoplasma* OTU DUBs.

| Name | Gene ID | BLAST search (human) | | DALI: PDB search | | DALI: AlphaFold search (human) | | DALI: AlphaFold search (all organisms) | | | |
|---|---|---|---|---|---|---|---|---|---|---|---|
| | | Top hit | Score | Top PDB hit | Z-score | Top human hit | Z-score | Top AlphaFold hit (gene) | Gene description | Organism | Z-score |
| TgOTUD1A | TGME49_207650 | OTUD1 | 44.7 | OTUD3 4BOU | 14.6 | OTUD1 | 14.9 | Tc00.1047053508961.10 | OTU domain–containing protein | *Trypanosoma cruzi* | 16.4 |
| TgOTUD1B | TGME49_237894 | OTUD1 | 46.2 | OTUD3 4BOU | 14.8 | OTUD1 | 14.9 | Tc00.1047053508961.10 | OTU domain–containing protein | *Trypanosoma cruzi* | 16.4 |
| TgOTUD1D | TGME49_237900 | OTUD1 | 47.8 | OTUD3 4BOU | 13.7 | OTUD1 | 14.9 | Tc00.1047053508961.10 | OTU domain–containing protein | *Trypanosoma cruzi* | 16.4 |
| TgOTUD1E | TGME49_210678 | OTUD1 | 48.1 | OTUD3 4BOU | 14.7 | OTUD1 | 14.9 | Tc00.1047053508961.10 | OTU domain–containing protein | *Trypanosoma cruzi* | 16.4 |
| TgOTUD2 | TGME49_277990 | OTUD2 | 158 | OTU1 4BOZ | 22.2 | OTUD2 | 22 | yod1 | OTU1 | *Dictyostelium discoideum* (slime mould) | 22.7 |
| TgOTUD3A | TGME49_258780 | OTUD3 | 59.3 | OTUD1 4BOP | 20.8 | OTUD1 | 19.6 | PF3D7_0923100 | OTU domain–containing protein, putative | *Plasmodium falciparum* | 23.1 |
| TgOTUD3B | TGME49_229710 | OTUD3 | 81.6 | OTUD3 4BOU | 18.4 | ALG13 | 16.8 | PF3D7_1031400.1 | OTU-like cysteine protease | *Plasmodium falciparum* | 19.8 |
| TgOTUD3C | TGME49_243430 | ALG13 | 70.09 | OTUD3 4BOU | 19.4 | OTUD5 | 17.8 | otud5a | OTU domain–containing protein 5A | *Danio rerio* (zebrafish) | 20.2 |
| TgOTUD6A | TGME49_271070 | OTUD6A | 34 | OTUD3 4BOU | 15.1 | OTUD6B | 13.2 | Os08g0506000 | Os08g0506000 protein | *Oryza sativa* subsp. japonica (rice) | 16.8 |
| TgOTUD6B | TGME49_243510 | OTUD6B | 105 | OTUD3 4BOU | 16.2 | OTUD6A | 22.1 | Otud6b | Ubiquitinyl hydrolase 1 | *Rattus norvegicus* (rat) | 23.8 |
| TgOTUB1 | TGME49_260510 | OTUB1 | 110 | OTUB1 4DDG | 29.9 | OTUB1 | 30.4 | OTUB1A | Ubiquitin thioesterase | *Danio rerio* (zebrafish) | 30.7 |
| TgOTU8 | TGME49_216440 | N/A | N/A | OTUD1 4BOP | 14.3 | OTUD1 | 13.3 | Tc00.1047053508961.10 | OTU domain–containing protein | *Trypanosoma cruzi* | 14.4 |
| TgOTU9 | TGME49_266500 | N/A | N/A | OTUD3 4BOU | 12.8 | OTUD1 | 12.2 | Duba (Dmel/CG6091) | Deubiquitinating enzyme A | *Drosophila melanogaster* (fruitfly) | 12.9 |
| TgOTU10 | TGME49_268690 | N/A | N/A | MelOTU 6w9r | 13.0 | OTUD2 | 12.9 | PF3D7_1350500 | Uncharacterised protein | *Plasmodium falciparum* | 20 |

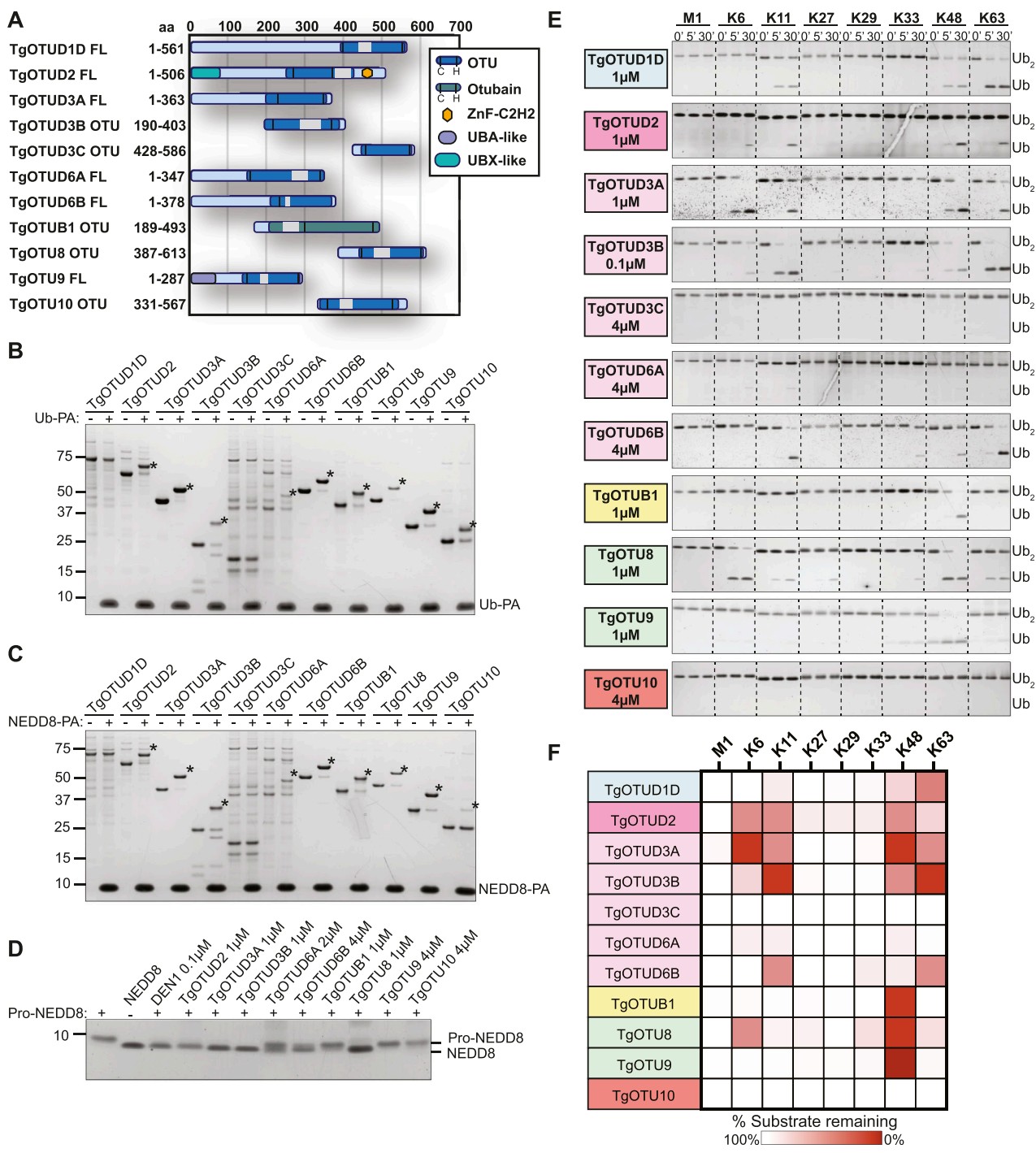

**Figure 3. Activity and linkage specificity of *Toxoplasma* OTU DUBs.**
**(A)** Constructs used in this study. Truncated constructs were used where full-length constructs could not be expressed or purified. **(A, B)** Reactivity of purified constructs from (A) against the activity-based probe Ub–propargylamine (Ub-PA, 1-h reactions). Asterisk (*): Ub-modified construct. **(A, C)** Reactivity of purified constructs from (A) against activity-based probe NEDD8-PA (1-h reactions). Asterisk (*): NEDD8-modified construct. **(D)** Cleavage of precursor pro-NEDD8 by TgOTU DUBs. Human DEN1 is used as a positive control. 1-h reactions were resolved on SDS–PAGE, and pro-NEDD8 cleavage was visualised by silver staining. **(E)** Activity of TgOTU DUBs against the eight diUb linkage types. **(A)** Purified constructs in (A) at indicated concentrations were incubated with each diUb linkage type (1.2 μM) and sampled at 0′, 5′, and 30′. Samples were resolved on SDS–PAGE, and diUb/monoUb was visualised by silver staining or Lumitein staining. **(E, F)** Qualitative heatmap summary based on densitometry of cleavage assays in (E). Data are representative of three independent experiments.

increase in molecular weight can be visualised as an upward shift on SDS–PAGE. All *Toxoplasma* OTU DUBs with the exception of TgOTUD1D and TgOTUD3C showed reactivity with Ub-PA, confirming

ubiquitin-binding activity for annotated OTU DUB family members and validating our predictions of ubiquitin-binding activity for TgOTUD6A, TgOTU9, and TgOTU10 (Fig 3B).

## Activity of *Toxoplasma* OTU DUBs against ubiquitin-like modifiers

We expanded these studies to other ubiquitin-like modifiers and used SUMO1-PA and NEDD8-PA activity-based probes. No *Toxoplasma* OTU DUB constructs were modified by SUMO1-PA (Fig S3B). All constructs modified by Ub-PA were also modified by NEDD8-PA (Fig 3C). Similarly, most human OTU enzymes can be modified by NEDD8-PA but will not hydrolyse a NEDD8-based substrate (Mevissen et al, 2013). To determine whether NEDD8 was a true substrate for *Toxoplasma* members, we tested OTU DUB activity against pro-NEDD8, which contains a 5-residue C-terminal peptide extension that is proteolytically cleaved by cellular NEDD8 proteases such as human DEN1, to generate functional NEDD8 (Kamitani et al, 1997). Interestingly and unlike human OTU DUBs, most *Toxoplasma* OTU DUBs processed pro-NEDD8 efficiently. TgOTUD2, TgOTUD3A, TgOTUD3B, and TgOTU8 appeared slightly more active as compared to TgOTUD6A, TgOTUD6B, and TgOTUB1. TgOTU9 and TgOTU10, despite being modified by NEDD8-PA, were unable to process pro-NEDD8 under these conditions (Fig 3D).

## Linkage specificity of *Toxoplasma* OTU DUBs against diubiquitin

A defining feature of OTU DUBs is their striking linkage specificity; human OTU enzymes evolved members targeting each of the eight ubiquitin linkage types, whereas bacterial effector OTU DUBs target a subset of chain types (Mevissen et al, 2013; Schubert et al, 2020). We hence tested the specificity of *Toxoplasma* OTU DUBs against the panel of eight diubiquitin (diUb) linkage types in time-course experiments using a gel-based cleavage assay (Licchesi et al, 2012; Mevissen et al, 2013). The results from this analysis are shown in Fig 3E and summarised in Fig 3F. Most of the *Toxoplasma* OTU DUBs preferentially cleaved Lys6-, Lys11-, Lys48-, and Lys63-linked diUb, but were much less or inactive against Lys27-, Lys29-, and Lys33-linked diUb. *Toxoplasma* does not contain an OTU DUB directed against M1-linked diUb (Fig 3E and F). Despite a lack of activity with Ub-PA, TgOTUD1D was able to cleave Lys11-, Lys48-, and Lys63-linked diUb (Fig 3E and F). TgOTUB1, like its human orthologue, showed specific activity towards Lys48 diUb (Mevissen et al, 2013). TgOTU9 also preferentially cleaved Lys48 diUb with additional background activity against Lys33 diUb.

TgOTUD3C and TgOTU10 were not active against any linkage type, whereas TgOTUD6A was minimally active against Lys6-, Lys11-, and Lys48-linked diUb (Fig 3E and F). To further interrogate the activity of these three OTU members, we tested their cleavage activity against a panel of triubiquitin (triUb) substrates. TgOTUD3C and TgOTU10 were unable to cleave any triUb substrates, and TgOTUD6A showed minimal cleavage activity against Lys6-linked triUb (Fig S3C). TgOTU10 showed low levels of diUb cleavage activity only with longer incubation times (Fig S3D).

The restricted linkage profile for *Toxoplasma* OTU DUBs for Lys6, Lys11, Lys48, and Lys63 linkages is a combination not seen in human enzymes but observed in viral and bacterial OTU DUBs (Mevissen et al, 2013; Dzimianski et al, 2019; Schubert et al, 2020).

## TgOTUD2 contains conserved accessory domains important for deubiquitinase function

We focussed our attention on TgOTUD2, which is similar to human OTUD2 and, unlike other TgOTU DUBs, contains accessory domains known to affect its linkage preferences (Mevissen et al, 2013). OTUD2 orthologues bind to the AAA+ ATPase/unfoldase p97/VCP and furnish it with ubiquitin linkage processing or editing capability (Rumpf & Jentsch, 2006; Messick et al, 2008; Ernst et al, 2009). The interaction is mediated by an N-terminal UBX domain in OTUD2 (Kim et al, 2014; Kim & Kim, 2014). In addition, most OTUD2 orthologues contain a C-terminal ZnF domain implicated in ubiquitin interactions (Fig 4A).

The TgOTUD2 OTU domain superimposes well with human OTUD2 (PDB IDs: 4BOS and 4BOZ; RMSD 0.783–0.881) (Fig 4B and C). In human OTUD2, the ZnF domain appears to broaden the ubiquitin linkage specificity profile of a Lys11-specific OTU domain, which prefers long Lys11-linked chains bound via an S2 Ub-binding site on the C-terminal α6-helix (Mevissen et al, 2013). TgOTUD2, however, lacks a similar S2 site (Fig 4C).

AlphaFold was able to predict structures for the UBX-like and ZnF domains of TgOTUD2 with high confidence (pLDDT > 70) (Fig 4D); however, a high predicted alignment error indicated the positioning of these domains was uncertain. To understand how these accessory domains affect activity and specificity, a series of TgOTUD2 deletion constructs were interrogated (Fig 4E). Interestingly, full-length TgOTUD2 has a preference for Lys6-, Lys11-, Lys48-, and Lys63-linked diubiquitin (Fig 4F). Removal of the ZnF domain from TgOTUD2 had no effect on linkage preference, whereas removal of the UBX-like domain resulted in a loss of activity against Lys63-linked diUb by the OTU-ZnF and OTU domain–only constructs (Fig 4F). This loss of Lys63 cleavage activity was also observed with triUb substrates (Fig 4G). These results stand in contrast to human OTUD2, in which removal of the ZnF domain reduces activity against several linkage types, whereas the UBX domain has little effect on linkage preference (Mevissen et al, 2013).

## AlphaFold predicts cryptic ubiquitin-like accessory domains with high confidence

AlphaFold prediction of full-length *Toxoplasma* OTU DUBs (Fig S4) confirmed the presence of highly disordered sequence at N- and C-termini for most enzymes, with one notable exception. For the diverse and apicomplexan-specific TgOTU9, AlphaFold predicted a cryptic ubiquitin-associated (UBA)–like domain in the enzyme's N-terminus (residues 1–64) (Fig 5A). This domain was not annotated by sequence-based methods (BLAST or hidden Markov model–based searches), but is conserved in two TgOTU9 orthologues in *Besnoitia besnoiti* and *Neospora caninum*, two close relatives of *Toxoplasma*. DALI indicated the highest similarity to the UBA domain of yeast Dsk2 (PDB ID: 4UN2; RMSD 0.847) (Michielssens et al, 2014; Holm, 2022) (Fig 5B). Interestingly, Dsk2 is a proteasome adaptor with a Lys48-specific UBA domain (Funakoshi et al, 2002), and TgOTU9 was identified to strongly prefer Lys48-linked diUb (Fig 3E and F).

We compared activity and specificity of full-length TgOTU9 with a construct comprising the OTU domain only (Fig 5C). Both constructs were able to bind the Ub-PA probe, indicating a functional OTU domain catalytic site (Fig 5D). Removal of the UBA-like domain completely abolished Lys48 activity, indicating this domain is essential for TgOTU9 activity (Fig 5E). This is yet another example of regulation of DUB activity and specificity via accessory domains,

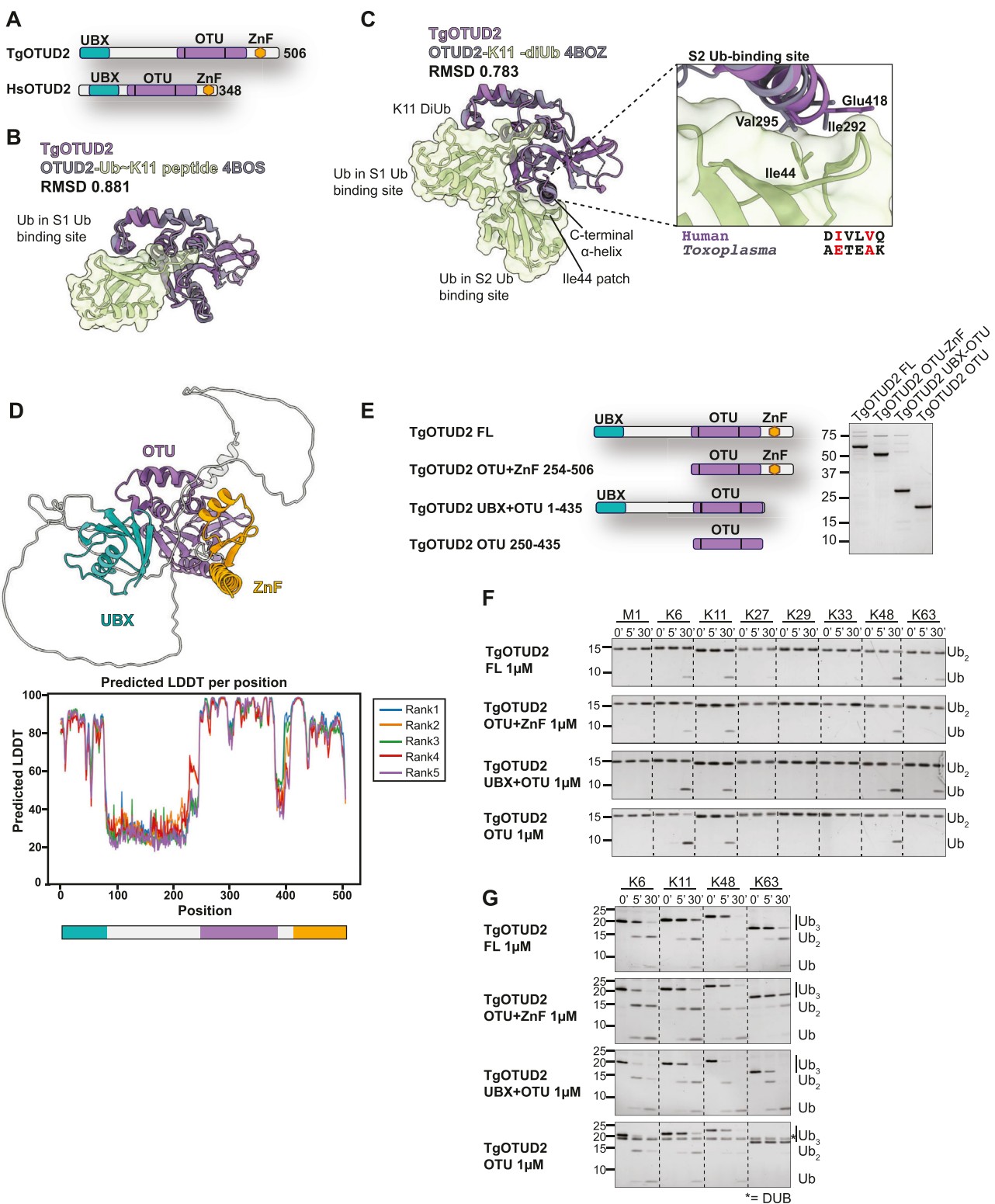

**Figure 4. Interrogation of TgOTUD2 accessory domains.**
**(A)** Superposition of TgOTUD2 with the human OTUD2 Ub~Lys11 peptide (PDB ID: 4BOS) (Mevissen et al, 2013). RMSD calculated with the ChimeraX Matchmaker tool on 136 pruned atom pairs of a total of 156 (Pettersen et al, 2021). **(B)** Left: superposition of TgOTUD2 with human OTUD2-Lys11-diUb (PDB ID: 4BOZ) (Mevissen et al, 2013). RMSD calculated with the ChimeraX Matchmaker tool on 138 pruned atom pairs of a total of 156 (Pettersen et al, 2021). Right: human OTUD2 S2 Ub-binding site comprises residues Ile292 and Val295 that interact with Ile44 of ubiquitin (Mevissen et al, 2013). Alignment shows these residues are not conserved in *Toxoplasma*. **(C)** Schematic of TgOTUD2 domain structure compared with human OTUD2. **(D)** Top: AlphaFold structural prediction of full-length TgOTUD2 highlighting the UBX domain (green) and the ZnF domain

and further exemplifies the power of AlphaFold to discover previously unannotated protein domains.

## Auxin-inducible degron rapidly induces degradation of TgOTUD3B and TgOTUD6B

An exciting advantage of *Toxoplasma* is the relative ease of genetic manipulation of its haploid genome, and well-established cell biology. As indicated in the Introduction section, genome-wide CRISPR screens have assigned essentiality scores for all *Toxoplasma* genes, indicating that several are important for its life cycle (Sidik et al, 2016). We sought to characterise two genes encoding TgOTUD3B and TgOTUD6B, with strong negative CRISPR phenotype scores of −3.46 and −2.99, respectively (Table 1) (Sidik et al, 2016). Rather than reverting to genetic knockouts, we used the mini-auxin-inducible degron (mAID) system, a rapid and reversible conditional knockdown approach to control protein levels at the post-translational level (Nishimura et al, 2009; Brown et al, 2018). OTU-encoding genes were modified with CRISPR/Cas9 to incorporate a HA-tagged mAID at the C-terminus, followed by a FLAG-tagged OsTIR1 expression cassette (Fig 6A). OsTIR1 recruits the conserved Skp1–Cullin–F-Box (SCF) ubiquitin ligase complex, facilitating ubiquitination and proteasomal degradation of mAID-tagged proteins in the presence of indole-3-acetic acid (IAA) (Nishimura et al, 2009; Brown et al, 2017, 2018).

Genetic modifications were confirmed by PCR, and Western blot detecting HA and FLAG tags (Figs 6B and S5). IAA treatment resulted in rapid degradation of TgOTUD3B and TgOTUD6B, as seen by near-complete loss of the HA signal after 60 min of treatment. IAA treatment had no effect on TIR1-FLAG expression (Fig 6B).

## TgOTUD knockdown parasites show differences in lytic stage growth

In order to assess the importance of TgOTUD3B and TgOTUD6B in intracellular growth, we performed plaque assays in which host cells were infected with a small number of parasites and were grown undisturbed for 7 d to allow for plaque formation. No changes in plaque number or size were observed after TgOTUD3B knockdown, despite prior prediction of this gene to be essential in tachyzoites (Fig 6C). TgOTUD6B knockdown had no effect on plaque number; however, plaque size was significantly reduced, indicating TgOTUD6B is important for parasite growth (Fig 6D and E). Microscopy probing for the HA tag of TgOTUD3B-AID and TgOTUD6B-AID parasites revealed a punctate perinuclear staining indicative of cytoplasmic localisation; however, given the TM domain in TgOTUD3B this staining could also be consistent with ER localisation (Fig 6F). To investigate the TgOTUD6B knockdown phenotype, we examined tachyzoite morphology by staining for inner membrane

complex protein 1 (IMC1), which is regulated by the cell cycle (Nishi et al, 2008). TgOTUD6B-AID parasites grown several cycles in the presence of IAA showed no morphological defects in the IMC (Fig 6G). Similarly, there were no defects observed in apicoplast localisation as indicated by staining of acyl carrier protein (ACP), an apicoplast marker (Waller et al, 1998). Tachyzoites appeared to be dividing normally, as shown by segregation of the apicoplast in dividing daughter cells (Fig 6H, white arrowheads). Further work will be necessary to determine the mechanism of the growth phenotype observed in TgOTUD6B knockdown parasites.

# Discussion

In this study, we have performed a comprehensive analysis of the OTU deubiquitinase family in *Toxoplasma*. Compared with other apicomplexans, this family is expanded in *Toxoplasma* and the closely related *Hammondia*, suggesting a functional relevance in late-branching coccidians. Structural predictions of the *Toxoplasma* OTU DUB family using AlphaFold confirm clear, recognisable OTU folds that align well with previously solved OTU domain crystal structures (Jumper et al, 2021; Mirdita et al, 2021 *Preprint*). Our attempts to crystallise full-length proteins were unsuccessful, likely because of the presence of large, disordered regions within most members of this family. Protein disorder is common in apicomplexan proteomes—more than 80% of *Toxoplasma* proteins are predicted to have intrinsically disordered regions of more than 30 residues, but the reason for this is unknown (see the Materials and Methods section) (Feng et al, 2006; Mohan et al, 2008). *Toxoplasma* OTU DUBs show diversity in length, domain organisation, and structural characteristics, particularly in the S1 Ub-binding site. Indeed, S1 Ub-binding sites in viral and bacterial OTU domains are highly diverse and are important in substrate specificity and positioning (Akutsu et al, 2011; James et al, 2011; Schubert et al, 2020). Several *Toxoplasma* OTU DUB members contain short insertions in this site, which may participate in substrate binding.

UBDs can be important for regulating OTU DUB function, working to restrict or broaden the specificity of an OTU domain (Mevissen et al, 2013). OTUD2 is highly conserved across eukaryotes, comprising both UBX and ZnF domains. In human OTUD2, the ZnF domain broadens the linkage specificity profile of a Lys11-specific OTU domain to target more chain types (Mevissen et al, 2013). In *Toxoplasma*, the UBX domain of TgOTUD2 is required for activity against Lys63-linked diUb, whereas the ZnF domain appears dispensable for activity. Through AlphaFold structural prediction of TgOTU9, we discovered an essential Lys48-specific UBA-like domain with no sequence similarities to known UBA domains. This domain shares structural similarity with yeast Dsk2, a known proteasome interactor that binds Lys48-linked polyUb (Funakoshi et al, 2002;

---

(orange). Bottom: predicted LDDT indicates confidence of a prediction at a given residue. Regions of low confidence (pLDDT < 50) are represented as unstructured loops (Source: ColabFold AlphaFold2_advanced notebook [Jumper et al, 2021; Mirdita et al, 2021 *Preprint*]). **(E)** Left: schematic of constructs generated to assess functionality of UBX and ZnF domains. Right: purified constructs resolved on a Coomassie-stained SDS–PAGE. **(F)** DiUb cleavage assays performed as in Fig 3E with different TgOTUD2 constructs: full-length; OTU + ZnF (missing UBX domain); UBX + OTU (missing ZnF domain); and OTU domain only. TgOTUD2 constructs (1 μM) were incubated with diUb at 1.2 μM. Removal of the UBX domain prevents activity against Lys63-linked diUb. **(G)** Cleavage activity against Lys-, Lys11-, Lys48-, and Lys63-linked triUb (1.2 μM) by different TgOTUD2 constructs (1 μM). TriUb cleavage assays performed as in Fig 3E. Removal of the UBX domain prevents activity against Lys63-linked triUb. Asterisk (*): TgOTUD2 OTU domain construct.

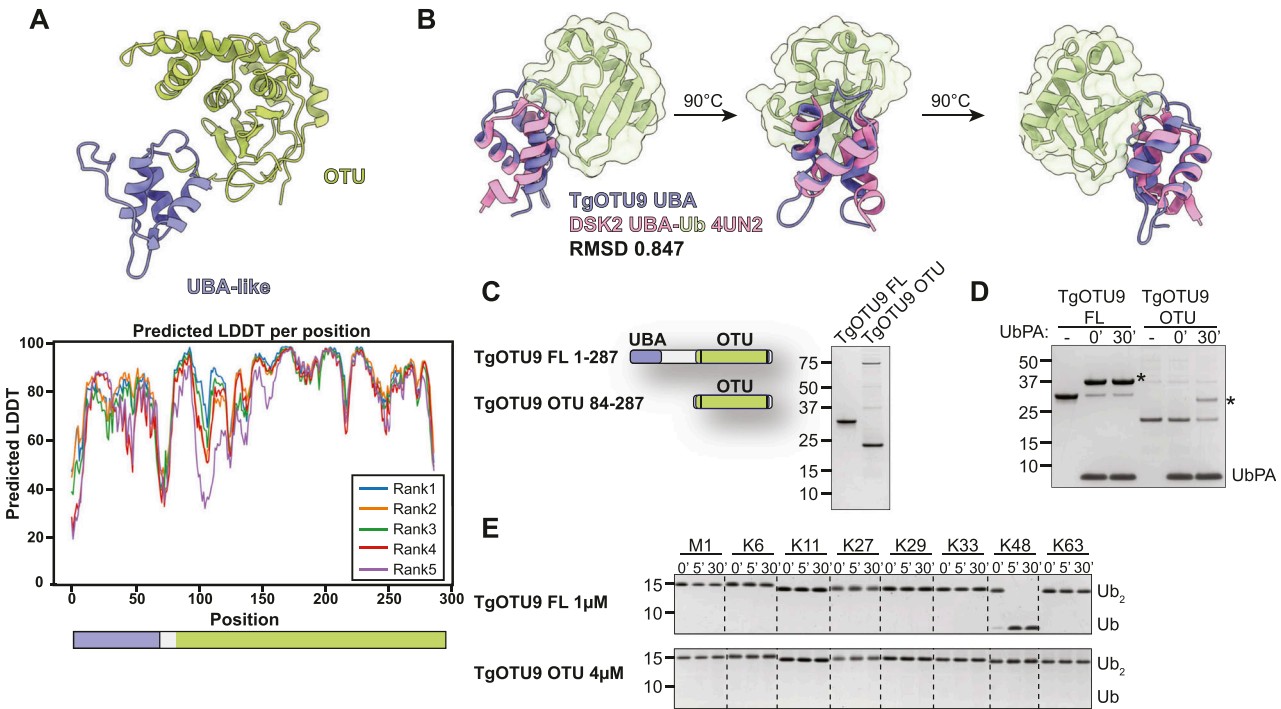

**Figure 5. TgOTU9 has a cryptic UBA-like domain that is critical for Lys48 activity.**
**(A)** Top: AlphaFold structural prediction of full-length TgOTU9 highlighting the UBA-like domain (lavender). Bottom: predicted IDDT indicates confidence of a prediction at a given residue (Source: ColabFold AlphaFold2_advanced notebook [Jumper et al, 2021; Mirdita et al, 2021 *Preprint*]). **(B)** Superposition of the TgOTU9 UBA domain with the UBA domain of Dsk2 (PDB ID: 4UN2) (Michielssens et al, 2014). RMSD calculated with the UCSF ChimeraX Matchmaker tool on 17 pruned atom pairs of a total of 40 (Pettersen et al, 2021). **(C)** Left: schematic of constructs generated to assess functionality of UBA-like domain. Right: purified constructs resolved on a Coomassie-stained SDS–PAGE. **(B, D)** Reactivity of constructs in (B) with Ub-PA activity-based probe at 0′ and 30′. Asterisk (*): Ub-modified construct. **(E)** DiUb cleavage assays performed as in Fig 3E with TgOTU9 full-length (1 $\mu$M) and OTU-only (4 $\mu$M) constructs. DiUb was used at 1.2 $\mu$M. Removal of the UBA-like domain prevents activity against Lys48-linked diUb.

Lowe et al, 2006). UBA domains are found in several deubiquitinases, autophagy receptors, and E3 ligases, and are known to be important for transducing signals via recognition of ubiquitinated substrates (Husnjak & Dikic, 2012). The UBA domain of Cezanne (human OTUD7B) is not only important for substrate recognition and binding, but is also required for its role in NF$\kappa$B inhibition, via binding to polyubiquitinated signalling proteins, which recruit the DUB to the tumour necrosis factor receptor (TNFR) complex (Ji et al, 2018; Mader et al, 2020). Further work will be required to determine whether the TgOTU9 UBA domain is also involved in recruitment or localisation, in addition to the catalytic activity of this DUB.

We have identified deubiquitinase activity and diUb linkage specificity profiles for nine of 11 OTU DUBs investigated, revealing a range of specificities for different linkage types. Most diUb cleavage profiles included Lys6, Lys11, Lys48, and Lys63 preferences, which are reminiscent of those seen in bacterial OTU enzymes (Schubert et al, 2020). The seven Lys residues of ubiquitin are conserved among all eukaryotes, and indeed, linkages through all Lys residues coexist in humans, yeast, and *Arabidopsis* (Peng et al, 2003; Xu et al, 2009; Berger et al, 2022). The only chain types previously detected in *Toxoplasma* are Lys11-, Lys48-, and Lys63-linked polyubiquitin chains (Dhara & Sinai, 2016). These chain types are also present in *Plasmodium* with the addition of Lys6 (Ponts et al, 2011). Our data are further evidence of the prominence of these chain types in

apicomplexan parasites. Lys6 is one of the least abundant chain types in humans, yet plays important roles in the DNA damage response (Elia et al, 2015) and mitophagy (Durcan et al, 2014; Cunningham et al, 2015; Gersch et al, 2017). In humans, OTUD3 is the only OTU DUB shown to efficiently cleave Lys6-linked diUb (Mevissen et al, 2013). We showed TgOTUD2, TgOTUD3A, TgOTUD3B, and TgOTU8 are all able to cleave Lys6-linked diUb, suggesting this chain type may have distinct roles in *Toxoplasma*. Conversely, Lys29 has been reported as one of the more abundant atypical chain types in humans (Swatek et al, 2019), but no Lys29-linked diUb cleavage was observed by any *Toxoplasma* OTU DUB.

In addition to being chain type–specific, OTU deubiquitinases also show varying specificities for ubiquitin and other ubiquitin-like modifiers (UBLs). Although the OTU DUB family is ubiquitin-specific in humans, in other organisms OTU DUBs have been found to hydrolyse other ubiquitin-like modifiers: several viral OTU domains cleave both Ub and interferon-stimulated gene 15 (ISG15) (Akutsu et al, 2011); bacterial OTU DUBs EschOTU and ceg23 can cleave NEDD8 and SUMO1, respectively (Schubert et al, 2020); and a *P. falciparum* OTU DUB can deconjugate Atg8 from membranes (Datta et al, 2017). Interestingly, we observed cleavage of human pro-NEDD8 by several *Toxoplasma* OTU DUB members. It should be noted that *Toxoplasma* NEDD8 only shares 47% identity with human NEDD8; therefore, whether *Tg*NEDD8 is a true substrate of *Toxoplasma* OTU enzymes will need to be determined.

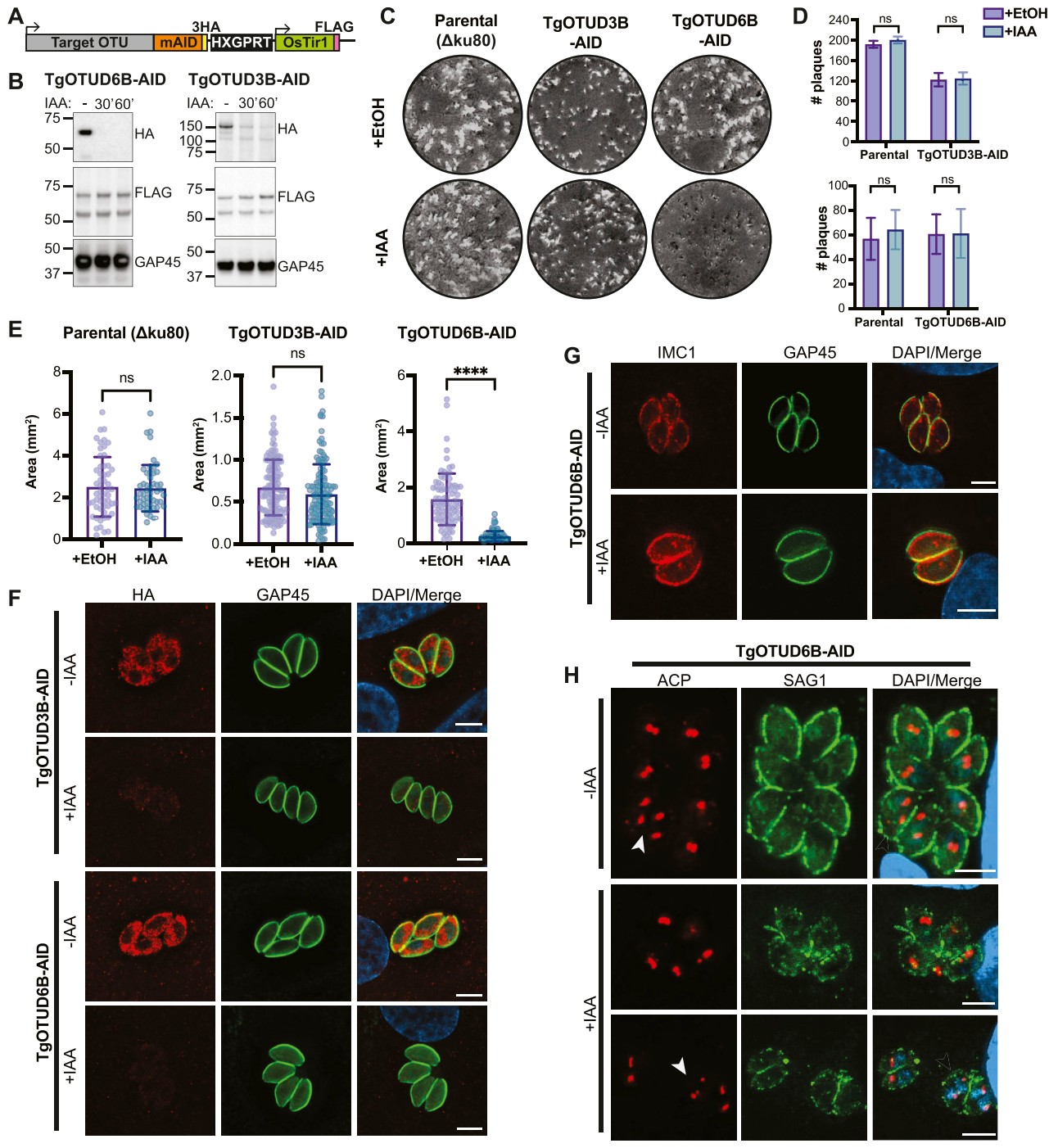

**Figure 6. Knockdown of TgOTUD3B and TgOTUD6B in *Toxoplasma* parasites.**
**(A)** Schematic representation of mAID-based conditional knockdown of TgOTUD3B and TgOTUD6B. Endogenous gene loci were modified to incorporate a 3′ mini-auxin-inducible degron (mAID), triple haemagglutinin tag (3HA), and HXGPRT selectable marker followed by the TIR1-FLAG expression cassette. Details of genetic strategy and validation of parasite lines are outlined in Fig S4. **(B)** Western blot of knockdown of TgOTUD3B-AID and TgOTUD6B-AID in intracellular tachyzoites after IAA treatment at 30′ and 60′ alongside vehicle treatment (EtOH). HA antibodies were used to detect TgOTUD6B-AID-HA or TgOTUD3B-AID-HA; FLAG antibodies were used to detect Tir1; and GAP45 was used as a loading control. **(C)** Plaque assays on confluent HFF monolayers infected with parental (Δku80), TgOTUD3B-AID, and TgOTUD6B-AID parasite lines in the presence of IAA (500 $\mu$M) or vehicle (EtOH). Monolayers were fixed and stained with crystal violet at 8 d post-infection. **(C, D)** Quantitation of plaque numbers in (C). $N = 3$ biological replicates; data are displayed as means ± SEM; ns, not significant. Statistical test: Mann–Whitney test. **(C, E)** Quantitation of plaque area (mm²) in (C). $N = 3$ biological replicates; data are displayed as means ± SEM; ns, not significant, ****$P < 0.0001$. Statistical test: Mann–Whitney test. **(F)** Immunofluorescence of TgOTUD3B-AID and TgOTUD6B-AID parasites after treatment with IAA or vehicle (EtOH) for 24 h. Parasites labelled with antibodies recognising HA and glideosome-associated protein 45 (GAP45), and counterstained with DAPI. Scale bars: 5 $\mu$M. **(G)** Inner membrane complex morphology after TgOTUD6B knockdown. Parasites treated with IAA or vehicle for three intracellular growth cycles before immunofluorescence. Parasites labelled with antibodies recognising inner membrane complex protein 1 (IMC1) and GAP45, and counterstained with DAPI. Scale bars: 5 $\mu$M. **(H)** Apicoplast morphology after TgOTUD6B knockdown. Parasites treated with IAA or vehicle for three intracellular growth cycles before immunofluorescence. Parasites labelled with antibodies recognising apicoplast marker acyl carrier protein and surface antigen 1 (SAG1), and counterstained with DAPI. White arrowheads: apicoplast segregation into daughter cells during cell division. Scale bars: 5 $\mu$M.

In this study, we were unable to determine substrates for TgO-TUD3C or TgOTU10. TgOTU10 could be modified by Ub-PA and NEDD8-PA; however, only weak activity with long incubation times was observed against diUb. Although resembling an OTU domain, TgOTU10 shows great diversity in its structure as compared to other family members, and does not align well with any known OTU domain structures. Similar sequences were found in related apicomplexans *P. falciparum* and *C. parvum*, and hyperLOPIT proteomics localised TgOTU10 to the apicoplast (Barylyuk et al, 2020). Together, these data suggest a conserved apicomplexan-specific OTU-like protein.

The apicoplast has its own endoplasmic reticulum–associated degradation–like ubiquitination system that has evolved to facilitate the import of nuclear-encoded proteins (Agrawal et al, 2013; Fellows et al, 2017). Successful translocation of proteins across the apicoplast periplastid membrane (one of four surrounding membranes) requires a plastid ubiquitin-like protein (PUBL), which is likely conjugated to proteins to permit import (Fellows et al, 2017). It is tempting to speculate that TgOTU10 might fulfil a deconjugation role during preprotein translocation. Another possibility is that TgOTU10 could act on Atg8, a UBL with a conserved role in autophagosome formation that also localises to the outer apicoplast membrane in *Toxoplasma* and *Plasmodium*, and plays a crucial, non-canonical role in apicoplast homeostasis and inheritance (Tomlins et al, 2013; Lévêque et al, 2015; Cheng et al, 2022). Given their localisation and divergent cellular functions, PUBL and Atg8 make plausible candidates for further investigation as potential TgOTU10 substrates.

To identify the importance of *Toxoplasma* OTU DUBs in parasite survival, we generated TgOTUD3B and TgOTUD6B AID-mediated knockdown lines. Parasites lacking TgOTUD3B showed no differences in growth, despite a predicted functional importance through a genome-wide CRISPR knockout screen (Sidik et al, 2016). This likely reflects differences in parasite culture conditions in this study versus those in a highly competitive screen format. With so many OTU-encoding genes in *Toxoplasma*, there may be functional redundancy among members, particularly those specific for Lys11, Lys48, and Lys63, such as TgOTUD3B and TgOTUD6B. TgOTUD6B knockdown parasites showed a significant reduction in plaque size, indicating that TgOTUD6B is required for parasite growth and survival; however, the precise mechanism remains to be elucidated. Future studies investigating the remaining OTU DUB members in *Toxoplasma*, particularly the less conserved TgOTU9 and TgOTU10, will likely provide further insights into the diversity and functions of this deubiquitinase family in apicomplexan parasites.

# Materials and Methods

### Prediction of apicomplexan OTU-encoding genes

A VEuPathDB search (https://veupathdb.org/) was performed within the indicated apicomplexan genomes for genes with predicted OTU domains (matching PFAM IDs: PF02238—OTU-like cysteine protease; and PF10275—Peptidase C65 Otubain) and their unannotated orthologues. Domain identification was cross-checked against PROSITE, and multiple sequence alignments were generated with T-Coffee to confirm the conservation of active site residues (catalytic Cys, His, and acidic) and to determine domain boundaries and identify insertions (Notredame et al, 2000; Sigrist et al, 2013). Apicomplexan OTU DUB candidates were further analysed through secondary structure prediction and OTU domain recognition with AlphaFold (Jumper et al, 2021). Data from ToxoDB (http://toxodb.org/toxo/) were compiled to further interrogate TgOTU DUBs using available genomic, transcriptomic, and proteomic data (Harb & Roos, 2020). Multiple sequence alignments were generated in ESPript (Supplemental Data 1.) (Robert & Gouet, 2014). Protein disorder was predicted using IUPred2A (Erdős & Dosztányi, 2020).

### TgOTU structural prediction and analysis

The structures of *Toxoplasma* OTU DUBs were predicted using the AlphaFold2 source code via the ColabFold notebook "Alpha-Fold2_Advanced" (Mirdita et al, 2021 *Preprint*). Predicted models with the highest LDDT scores are shown. Surface conservation analysis was carried out using ConSurf based on multiple sequence alignments in Supplemental Data 1. (Ashkenazy et al, 2016). The DALI protein structure comparison server was used to compare AlphaFold structural predictions of TgOTU DUBs with PDB structures or AlphaFold structures and identify structural orthologues (Holm, 2022). UCSF ChimeraX was used for all structural analysis and figure generation. Structural superpositions were generated with the Matchmaker tool using the Needleman–Wunsch sequence alignment and iterated fitting by pruning long atom pairs with an iteration cut-off distance of 2.0 Å. RMSD values reflect pruned atom pairs (Pettersen et al, 2021). A full list of RMSDs, atom pairs, and sequence alignment scores is outlined in Table S1.

### Annotation of the OTU complement in *Toxoplasma gondii*

Gene lists from previous analyses of the *Toxoplasma gondii* OTU complement were compiled and compared with genes annotated as OTU family cysteine proteases or ubiquitin thioesterase otubain–like family proteins on ToxoDB (Table S2). The *Toxoplasma* genome contains 14 annotated OTU family cysteine proteases, plus a 15th otubain subfamily protein (Harb & Roos, 2020). Four annotated OTUs with near-identical sequence (TGME49_323200, TGME49_323600, TGME49_323700, and TGME49_323800) were excluded based on a lack of assigned genomic location and expression data, potentially representing a misannotation in the genome. This yielded a final list of 14 OTU members in *Toxoplasma gondii*.

### Naming of TgOTU family members

Several *Toxoplasma* OTU domain–encoding genes had previously been named (Dhara & Sinai, 2016). One of these genes included TGME49_323200 (TgOTUD1C), which we excluded from our analysis. We also identified two other OTUD1 orthologues: TGME49_210678 and TGME49_237900, which we named TgOTUD1D and TgOTUD1E to avoid confusion over assignment of TgOTUD1C. We have updated naming for TGME49_271070 and TGME49_243510 (previously TgOTU7 and TgOTUD5) to TgOTUD6A and TgOTUD6B to reflect their sequence and structural similarity with human OTUD6A and OTUD6B. In

concordance with a naming convention, TGME49_268690 was named TgOTU10 (Table S2).

## Cloning

*Toxoplasma* OTU sequences were amplified from cDNA or purchased as synthetic DNA fragments (gBlocks) codon-optimised for bacterial expression (Integrated DNA Technologies, Inc.). Constructs for bacterial expression were cloned into the pOPINK vector (Berrow et al, 2007) using In-Fusion Cloning (Takara Bio) incorporating an N-terminal GST tag and a 3C cleavage site. Primers used are listed in Table S3.

## Protein expression and purification

GST-tagged OTU constructs were transformed into *E. coli* Rosetta2 (DE3) pLacI cells (Novagen), and cells were grown at 37°C in 2×YT medium until an $OD_{600}$ of 0.6–0.8 was reached. Protein expression was induced by the addition of 200 $\mu$M IPTG, and cultures were incubated overnight at 18°C. Cells were harvested by centrifugation at 5,000$g$ for 15 min at 4°C and frozen at –80°C. Cells were lysed by sonication in purification buffer (25 mM Tris [pH 8.5], 150 mM NaCl, 10% [vol/vol] glycerol, and 1 mM DTT) supplemented with EDTA-free protease inhibitor cocktail tablets (Roche), lysozyme, and DNase I (Sigma-Aldrich). Lysates were clarified by centrifugation at 50,000$g$ for 30 min at 4°C, and the supernatant was incubated with Glutathione Sepharose 4B resin (Cytiva). After washing with purification buffer, the resin was incubated with GST-3C PreScission Protease overnight to cleave off the GST tag. Cleaved proteins were concentrated and purified by size-exclusion chromatography (SEC) using HiLoad 16/600 Superdex columns in SEC buffer (20 mM Tris [pH 8.5], 150 mM NaCl, and 1 mM TCEP). Fractions containing pure protein were pooled, concentrated, flash-frozen in liquid nitrogen, and stored at –80°C.

## Modification of OTU DUBs by activity-based probes

Ub-, NEDD8-, and SUMO–propargylamine probes were generated according to Ekkebus et al (2013). Activity-based probe reactions were performed as described in Mevissen et al (2013). Briefly, TgOTU DUBs were prepared at 1 mg/ml in 5 mM DTT/PBS and mixed 1:1 with Ub-PA, NEDD8-PA, or SUMO1-PA at 1 mg/ml. Reactions were incubated at 37°C for the indicated times before addition of 4X SDS sample buffer to stop the reaction. 5 $\mu$l samples were resolved by SDS–PAGE and visualised using InstantBlue Coomassie Protein Stain (Abcam).

## NEDD8 hydrolase assay

The NEDD8 hydrolase assay was performed as previously described (Wu et al, 2003). TgOTU DUBs at the indicated concentrations were added to 0.5 $\mu$M pro-NEDD8 in 25 mM Tris, 150 mM NaCl, and 10 mM DTT, pH 7.4, and incubated at 37°C for 1 h. DEN1 (0.1 $\mu$M) was used as a positive control. Reaction products were resolved by SDS–PAGE and visualised by silver staining (Bio-Rad).

## In vitro DUB assays

In vitro ubiquitin linkage specificity assays were performed as described in Licchesi et al (2012). Briefly, 2X reaction stocks were prepared for each diUb at 2.5 $\mu$M in 25 mM Tris and 150 mM NaCl, pH 7.4. TgOTU DUBs were prepared at 2X concentration in 25 mM Tris, 150 mM NaCl, and 10 mM DTT, pH 7.4. Initial pilot experiments identified optimal DUB concentrations for robust diUb cleavage. Reactions were performed by mixing 1:1 diUb and DUB followed by incubation at 37°C. 9 $\mu$l reaction samples were taken at the indicated times and quenched with 3 $\mu$l 4X SDS sample buffer. Samples were resolved by SDS–PAGE and visualised with the Silver Stain Plus kit (Bio-Rad) or One-Step Lumitein UV Protein Gel Stain (Biotium).

## *Toxoplasma* in vitro culture

*Toxoplasma* parental lines used in this study include type I RH or type II Pruginaud (Pru) strains, both expressing Δku80 (RHΔku80 and PruΔku80) (Huynh & Carruthers, 2009; Fox et al, 2011). *Toxoplasma* tachyzoites were cultured under standard conditions in primary human foreskin fibroblasts (HFFs; American Tissue Culture Collection [ATCC]). HFFs were grown in DMEM supplemented with 10% heat-inactivated Newborn Calf Serum (GE Healthcare). Upon infection of HFFs with tachyzoites, media were changed to DME supplemented with 1% foetal calf serum (GE Healthcare). Cells were grown in humidified incubators at 37°C/10% $CO_2$.

## DNA, plasmids, and transfection

Modification of candidate genes was performed using CRISPR/Cas9, using an established protocol for use in *Toxoplasma* (Sidik et al, 2016; Shen et al, 2017). Briefly, a plasmid expressing sgRNA was cotransfected with targeted amplicon DNA to modify the 3′ end of the gene incorporating mAID-3XHA followed by a triple FLAG-tagged TIR1 expression cassette *in cis* (Brown et al, 2017, 2018; Lee et al, 2021 *Preprint*). CHOPCHOP (Labun et al, 2019) was used to design CRISPR guide sequences (Table S4), which were cloned into the pU6-Universal::mCherry plasmid (Sidik et al, 2016) using Q5 site-directed mutagenesis (New England Biolabs). Amplicon DNA was obtained via PCR of mAID-3XHA-HXGPRT-TIR1 (Lee et al, 2021 *Preprint*) using primers with 40 bp of homology to the insertion site to facilitate double homologous recombination (primers listed in Table S5).

Transfections were performed with the Amaxa 4D-Nucleofector (Lonza), using $2 \times 10^6$ tachyzoites and 10 $\mu$g each of plasmid and amplicon DNA resuspended in 20 $\mu$l P3 solution (Lonza). Nucleofection proceeded using program F1-115. Transfected parasites were placed on drug selection with mycophenolic acid (25 $\mu$g/ml) and xanthene (50 $\mu$g/ml) for three growth cycles before cloning out by serial dilution. Positive clones were identified by DNA sequencing (primers listed in Table S6), immunofluorescent imaging, and Western blot.

## Auxin-induced depletion of mAID-tagged proteins

Depletion of mAID-tagged proteins using IAA was performed as described in Brown et al (2017). Briefly, IAA was dissolved in 100% EtOH to a stock concentration of 500 mM and used at 1:1,000 to treat

parasites at a final concentration of 500 $\mu$M. Vehicle treatments comprised an equivalent volume of 100% EtOH.

## Western blotting

Immunoblot samples were pelleted and lysed for 30 min at 4°C in 1% (vol/vol) Triton X-100 (Sigma-Aldrich) and 1 mM MgCl$_2$ in PBS (Gibco) supplemented with 1× cOmplete protease inhibitors (Roche) and 0.2% (vol/vol) Benzonase (Merck). An equal volume of 2X reducing SDS sample buffer was added, and 15 $\mu$l samples were resolved on NuPAGE 4–12% Bis–Tris gels (Invitrogen) in 1X MOPS (Thermo Fisher Scientific). Proteins were transferred to nitrocellulose membranes using Trans-Blot Turbo Transfer System (Bio-Rad). Membranes were blocked in 5% (wt/vol) milk powder in Tris-buffered saline containing 0.1% (vol/vol) Tween-20 (TBS-T) for 1 h, then incubated with primary antibodies diluted in 5% milk/TBS-T overnight at 4°C. Primary antibodies used were as follows: rabbit $\alpha$-HA (3F10, 1:1,000, Cat# 11867423001, RRID:AB_390918; Roche); mouse $\alpha$-FLAG (1:1,000, Cat# F1804, RRID:AB_262044; Sigma-Aldrich); and rabbit $\alpha$-GAP45 (1:1,000) (Gaskins et al, 2004) (Table S7). Membranes were washed with TBS-T and incubated with HRP-conjugated secondary antibodies (Southern Biotech; see Table S8) diluted in 5% milk/TBS-T at 1:1,000 for 1 h before washing with TBS-T. Clarity Western ECL Substrate (Bio-Rad) was applied directly to membranes, and proteins were visualised using the ChemiDoc Gel Imaging System (Bio-Rad).

## Plaque assays

Plaque assays were performed by inoculating 100 tachyzoites onto confluent monolayers of HFFs grown in six-well plates (Sigma-Aldrich). Cells were left undisturbed for 7–9 d, and monolayers were then fixed in 80% EtOH and stained with violet (Sigma-Aldrich). Plaque number and size were quantitated using ImageJ software and analysed in Prism (GraphPad). Samples were tested for normal distribution before statistical testing.

## IF microscopy

Tachyzoites treated with IAA or vehicle for three growth cycles were inoculated onto coverslips with confluent monolayers of HFFs and grown for 24 h. Cells were fixed in 4% (vol/vol) paraformaldehyde/PBS (Sigma-Aldrich) for 10 min and permeabilised in 0.1% (vol/vol) Triton X-100/PBS before blocking in 3% (wt/vol) BSA/PBS (Sigma-Aldrich) for 1 h. Cells were incubated with primary antibodies used at 1:1,000 in 3% BSA/PBS for 1 h. Primary antibodies used were as follows: rat $\alpha$-HA 3F10 (1:1,000, Cat# 11867423001, RRID:AB_390918; Roche); mouse $\alpha$-SAG1 DG52 (Burg et al, 1988); rabbit $\alpha$-GAP45 (1:1,000) (Gaskins et al, 2004); rabbit $\alpha$-ACP (Waller et al, 1998); and rabbit $\alpha$-IMC1 (Ward & Carey, 1999) (Table S7). Cells were washed 3X in PBS and probed for 1 h with Alexa Fluor–conjugated secondary antibodies (Thermo Fisher Scientific; see Table S8 for full list), plus 5 $\mu$g/ml DAPI used at 1:1,000 in 3% BSA/PBS. Coverslips were washed and mounted onto glass microscope slides with Vectashield (Vector Labs). Parasites were imaged on a Zeiss Live Cell Axio Observer widefield microscope. Images were processed in ImageJ (Schneider et al, 2012).

# Data Availability

All reagents, materials, and raw data are available upon reasonable request from the corresponding authors.

# Supplementary Information

# Acknowledgements

We thank the members of the Komander, Lechtenberg, Tonkin, and Cowman laboratories at WEHI for reagents, advice, and comments on the article; and the staff at the WEHI Centre for Dynamic Imaging for providing facilities and support. D Komander is recipient of NHMRC Investigator Grant (GNT1178122). CJ Tonkin is the recipient of an NHMRC Ideas Grant (GNT1183496). M-L Wilde is a recipient of an Australian Government Research Training Program Scholarship. Research was further supported by an NHMRC Independent Research Institutes Infrastructure Support Scheme Grant (361646) and Victorian State Government Operational Infrastructure Support Grant. Crystallographic data collection was undertaken using the MX2 beamline at the Australian Synchrotron, part of ANSTO, which made use of the Australian Cancer Research Foundation (ACRF) detector.

## Author Contributions

M-L Wilde: conceptualisation, data curation, formal analysis, validation, investigation, visualisation, methodology, and writing—original draft, review, and editing.
U Ruparel: resources, data curation, formal analysis, investigation, visualisation, and methodology.
T Klemm: resources, methodology, and writing—review and editing.
VV Lee: resources and writing—review and editing.
DJ Calleja: resources, methodology, and writing—review and editing.
D Komander: conceptualisation, resources, supervision, funding acquisition, and writing—review and editing.
CJ Tonkin: conceptualisation, resources, supervision, funding acquisition, and writing—review and editing.

## Conflict of Interest Statement

D Komander is a founder and shareholder of Accrue Therapeutics, and serves on the Scientific Advisory Board of BioTheryX, Inc. The other authors declare no competing interests.

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
