## [Reviewer comments · Life Science Alliance]

Life Science Alliance

Characterisation of the OTU domain deubiquitinase complement of *Toxoplasma gondii*

Mary-Lou Wilde, Ushma Ruparel, Theresa Klemm, V Vern Lee, Dale Calleja, David Komander, and Christopher Tonkin
DOI: <https://doi.org/10.26508/lsa.202201710>

Corresponding author(s): David Komander, Walter and Eliza Hall Institute of Medical Research and Christopher Tonkin, Walter and Eliza Hall Institute of Medical Research

Review Timeline:

Submission Date:	2022-09-05
Editorial Decision:	2022-09-30
Revision Received:	2023-02-23
Editorial Decision:	2023-03-14
Revision Received:	2023-03-15
Accepted:	2023-03-15

Scientific Editor: Novella Guidi

Transaction Report:

September 30, 2022

Re: Life Science Alliance manuscript #LSA-2022-01710-T

Prof. David Komander
The Walter and Eliza Hall Institute of Medical Research
Ubiquitin signalling
1G Royal Parade
Parkville
Melbourne, Victoria 3052
Australia

Dear Dr. Komander,

Thank you for submitting your manuscript entitled "Characterisation of the OTU domain deubiquitinase complement of *Toxoplasma gondii*" to Life Science Alliance. The manuscript was assessed by expert reviewers, whose comments are appended to this letter. We invite you to submit a revised manuscript addressing the Reviewer comments.

Thank you for this interesting contribution to Life Science Alliance. We are looking forward to receiving your revised manuscript.

Sincerely,

B. MANUSCRIPT ORGANIZATION AND FORMATTING:

Reviewer #1 (Comments to the Authors (Required)):

In this study, Wilde and colleagues searched the ovarian tumor domain-containing (OTU) family of deubiquitinases (DUBs) in *Toxoplasma*, and characterized the properties and functions by bioinformatics and biochemical methods. Structural analysis using AlphaFold predictions in combination with DALI protein structure comparison revealed that orthologues of *Toxoplasma* OTU DUBs are found in another apicomplexan parasite, *Plasmodium*, succeeding development of a superior functional framework for annotation of enzyme classes in less studied organisms such as *Toxoplasma*. The authors, in turn, biochemically analyzed the OTU DUBs using the recombinant proteins, searched the specificity to ubiquitin and ubiquitin-like modifiers and yielded comprehensive linkage specificity of *Toxoplasma* DUBs against M1 and K6, K11, K27, K29, K33, K48 and K63. Then, the authors selected OTUD2 and OTU9, and found that accessory domains of OTUD2 and cryptic UBA-domain of OTU9 are important for the biochemical functions. Moreover, OTUD3 and OTUD6B were further biologically analyzed by generating auxin-inducible degron (AID) conjugating knockin mutants, and found that OTUD6B is important for the parasite growth. Overall, this reviewer considers that the current study is informative in terms of the bioinformatic approaches and biochemical properties of *Toxoplasma* OTU DUBs, but feels that the biological significance and relevance are less impressive. (See below)

(Major points)

1) The authors should generate the AID knockin system (or conventional knockout) of other OTU DUBs as well as OTUD3 and OTUD6B to analyze the biological significance in vitro and in vivo. OTUD3-deficient *Toxoplasma* showed normal growth in vitro, however, are OTUD3-deficient parasites able to grow in mice (in vivo)? Not only bioinformatics and biochemical analyzes but also biological significance (parasite growth in vitro and virulence in vivo) of all of the *Toxoplasma* OTU DUBs should be tested at minimum.

2) OTUD6B-deficient *Toxoplasma* exhibited growth retardation in vitro (Figure 6). OTUD6B displayed specificity on K11 and K63 Ub linkages (Figure 3F). What *Toxoplasma* proteins are affected by the lack of OTUD6B? The authors should explore the substrate of OTUD6B by comparing wild-type and OTUD6B-deficient parasites using mass spectrometry. Among the list of potential substrates, the authors should discuss as to which protein(s) regulated by OTUD6B are important for the in vitro phenotype.

(minor points)

3) Where is Figure 2D?

4) Table S1 was not reader friendly. The table should include information of CRISPR fitness score and HyperLOPIT, which make reader understood at a glance.

Reviewer #3 (Comments to the Authors (Required)):

The manuscript describes a comprehensive approach to identifying and characterizing OTU deubiquitinases of *Toxoplasma*. Several genes expression proteins with DUB activity were identified and characterized, including Ub linkage specificity, activity against UBLs, and functional dissection of accessory domains on some of the DUBs. The work provides a excellent foundation for further studies on OTU DUBs of *Toxoplasma*.

They first identified and listed the possible proteins in OTU DUB family of *Toxoplasma*, then predicted their 3-dimensional structures by using AlphaFold2. Based on this prediction, they compared these with OTU DUBs in other organisms, and designed constructs for in vitro studies. This paper makes excellent use of AlphaFold as a powerful data mining tool.

They identified whether catalytic sites of the cloned DUBs are active by using the probe Ub-HA with their proteins in vitro. Most of them showed ubiquitin binding, and interestingly showed pro-NEDD8 cleavage. In addition, they identified the DUBs' lysine linkage specificity of diUb. Some of them showed K6, K11, K48 and K63 specific cleavage, while TgOTU10 didn't show any of diUb cleavage even though it binds to Ub.

TgOTUD2, which has Zinc finger domain and UBX domain, resembles human OTU2. They attempted to identify the roles of

these accessory domain, however only UBX domain seems responsible for substrate recognition of K63-linked diUb. Contrary to human OTU2, the ZF of TgOTU2 doesn't seem to affect substrate recognition.

TgOTU9 has a UBA-like accessory domain, which structure is similar to UBA domain of Dsk2 (which gives K48 specificity) in yeast, and the deletion of this domain abolished the K48 specific deubiquitinase activity of TgOTU9.

Finally, they attempted to knock down these DUBs (especially for TgOTU3B and TgOTU6B, based on the CRISPR screen score of previous report) in an AID model to identify their role in vivo. Suppression of TgOUT3B did not show effects on plaque growth, however TgOTU6B deletion led decrease in plaque size.

Overall, the paper is nicely written and provides important new contributions to parasitology Ub-biology.

General Comments

The manuscript is well written with lots of background and relevant topics related to the OTU domain deubiquitinases of *Toxoplasma*. The Figures are really nice.

There are several typos and mislabeled units however (Line 182, 493, Figure 5B and legends of Figure 6 and possibly more).

Table 1 is missing from in the manuscript, which is expected to show the CRISPR scores. This is one of the most important parts of this research, because they should show how they selected their candidates.

For Figure EV1 and manuscript, need to state how RMSDs were calculated (which atoms and regions were used for calculation). RMSD values appear quite small, which could be due to how much of each model was used in the calculations.

Knockdown of TgOTU3B in vivo results showed different result than expected, even though it was based on based on the previous screening results. Regarding this, CRISPR score value should be mentioned together (maybe it's on Table 1??). Suggest showing value of in vivo experiment data (means and errors etc) in the manuscript or figure to more clearly demonstrate differences.

Most of the tables are not completed and not properly titled. This needs to be addressed.

Why was the linkage specificity limited to di-ubiquitin (diUb) substrates? Perhaps some of the DUBs cannot recognize di-Ub and thus some activities have been missed? This could also explain lack of activity toward Ub-propargylamine by some of the enzymes as well perhaps?

Minor points:

Line 40: 'and' NEDD8-based substrates? Replace 'but'with 'and'??

Line 48; *Toxoplasma* (should be in italics)

Line 71; The abbreviation for NFκB is missing

Line 104 and 108; Abbreviations for some terms are missing

Line 151 and 152; line spacing?

Line 158: Which software you used to generate the phylogenetic tree? (Figure 1A)

Line 165; Which software or method you used to interpret the data? Data source?

Line 179, 185; Table 1 is not included in the supplementary document?

Line 179-181; Line spacing?

Line 215, 548; Appendix data S1 is not included in the section.

Line 276; How many constructs of full length and catalytic variants?

Line 391; Abbreviation for SCF?

Line 458; Abbreviation for TNRF?

Line 487; define ISG.

Line 493; Typo error. Will need to be determined

Line 534; Line spacing?

Line 596; Table S2 is not included (primers?)

Line 637; Appendix Table S4 is not included

Line 645; Appendix Table S5 is not included

Line 667; Appendix Table S7 is not included

Line 689; Appendix Table S6 is not included

We thank the reviewers for taking their time to critically review our manuscript and provide constructive criticism. Below you will find comments from reviewers in red and our responses in black. Before doing so, it is important to point out that this paper was the result of the first author – Mary-Louise Wilde’s PhD, which she has now completed and passed. Dr Wilde has now moved onto a postdoctoral position in a new lab working on a completely new topic. We therefore had limited capacity to perform additional experiments, especially the extensive list the reviewer 1 asks of us. However, we felt like it was important to attempt some suggested experiments which we outline below. This was performed by newly added author Ms Ushma Ruparel who undertook the mouse virulence testing.

Reviewer #1 :

In this study, Wilde and colleagues searched the ovarian tumor domain-containing (OTU) family of deubiquitinases (DUBs) in *Toxoplasma*, and characterized the properties and functions by bioinformatics and biochemical methods. Structural analysis using AlphaFold predictions in combination with DALI protein structure comparison revealed that orthologues of *Toxoplasma* OTU DUBs are found in another apicomplexan parasite, *Plasmodium*, succeeding development of a superior functional framework for annotation of enzyme classes in less studied organisms such as *Toxoplasma*. The authors, in turn, biochemically analyzed the OTU DUBs using the recombinant proteins, searched the specificity to ubiquitin and ubiquitin-like modifiers and yielded comprehensive linkage specificity of *Toxoplasma* DUBs against M1 and K6, K11, K27, K29, K33, K48 and K63. Then, the authors selected OTUD2 and OTU9, and found that accessory domains of OTUD2 and cryptic UBA-domain of OTU9 are important for the biochemical functions. Moreover, OTUD3 and OTUD6B were further biologically analyzed by generating auxin-inducible degron (AID) conjugating knockin mutants, and found that OTUD6B is important for the parasite growth. Overall, this reviewer considers that the current study is informative in terms of the bioinformatic approaches and biochemical properties of *Toxoplasma* OTU DUBs, but feels that the biological significance and relevance are less impressive. (See below)

(Major points)

1) The authors should generate the AID knockin system (or conventional knockout) of other OTU DUBs as well as OTUD3 and OTUD6B to analyze the biological significance in vitro and in vivo. OTUD3-deficient *Toxoplasma* showed normal growth in vitro, however, are OTUD3-deficient parasites able to grow in mice (in vivo)? Not only bioinformatics and biochemical analyzes but also biological significance (parasite growth in vitro and virulence in vivo) of all of the *Toxoplasma* OTU DUBs should be tested at minimum.

Generation of mutants of all OTUs:

To generate knockout lines for all OTU DUB members and characterize them would take at least a year to complete. We also can see little overall benefit to the manuscript after performing these experiments without further interrogation of OTU DUB function. This would take additional time as we would want to run each mutant through standard assays in the lab to measure growth, understand which stage of parasite asexual cycle is affected

(ie invasion, replication, host cell egress) as well as identify and characterize potential substrates using proteomics and downstream functional analysis. Whilst we agree this would be interesting to it's a huge amount of work and is well beyond the scope of this study.

Virulence testing in mice:

As requested, we have now performed virulence testing of our *Toxoplasma* OTU mutants in mice. To do these experiments we grew our parasite strains in tissue culture, and infected animals intraperitoneally with inoculums listed below. We infected 6 mice per condition, including two groups infected with parental strain to control for any effect of indol acetic acid (IAA) (which promotes the depletion of AID-tagged protein) on infection kinetics (36 mice in experiment 1 and 12 mice in experiment 2 = 48 mice total).

For IAA dosing we relied on the one paper that has reported the use of the IAA/AID system t in mice infected with *Toxoplasma* (Brown et al, PMID: 30449726). This involved giving animals IAA in drinking water (ad libitum) and administered with additional IAA by oral gavage once daily. Mice were then weighed daily and monitored for time taken to reach ethical endpoint (moribund), both of which are measures of virulence of the infecting strain.

Unfortunately, our experiments did not provide any interesting findings (Fig 1). Before describing these, it is important to make clear the limitations of our experiments:

1. The AID (Auxin Inducible Degron) conditional protein depletion system is not extensively used in *Toxoplasma*-infected mice. This system requires that orally dosed Indol Acetic Acid (IAA) to be bioavailable and in high enough concentration across tissue types to permit knockdown of the AID fusion protein in *Toxoplasma*. There has been one publication showing positive results from this (albeit of a very fitness conferring gene, meaning that any depletion would likely lead to changes in virulence) and so we thought it was still worth attempting (Brown et al Cell Host and Microbe, PMID: 30449726).
2. The conditional knockout strains that we have on hand are in different genetic backgrounds. OTUD3B-AID is in 'Pru', which is a 'type II' strain and of intermediate virulence (we infected with 2000 parasites per mouse) and widely used in mouse studies. OTUD6B-AID is in 'RH' which is an extremely virulent 'type I' strain. Mice can only be infected with very low doses of RH (we used 15 parasites per mouse). Therefore, any small differences in virulence can be very hard to see in this strain. Given we saw a small growth defect upon depletion of OTUD6B in tissue culture we thought it would be worthwhile to test virulence in mice.
3. We only have one mouse strain setup for *Toxoplasma* infection (C57BL/6). Different breeds of mice do have different susceptibilities to *Toxoplasma* strains and this may affect the results.

Results: There was no effect on mouse weight or time to death upon infection with parental Pru strain and administration of IAA, showing that this compound does not have any measurable impact on virulence in our mouse model (Fig 1Ai).

We saw no difference in weight and time to death of animals infected with Pru OTUD3B-AID with or without IAA treatment (Fig 1Bi and ii), suggesting no change in virulence upon knockdown of OTUD3B. This is in line with no difference observed in *Toxoplasma* growth in tissue culture upon depletion of OTUD3B-AID with IAA.

We performed two experiments of mice infected with OTUD6B-AID (RH strain). In Experiment 1 upon treatment with IAA (knockdown of OTUD6B-AID) we saw a small, but measurable drop in weight compared to vehicle only, which is the reverse of what we would expect if OTUD6B depletion reduces virulence (Fig 1Bi). However, we observed increased survival of OTUD6B-AID-infected mice treated with IAA (inducing protein knockdown). This difference was statistically significant, suggesting that loss of OTUD6B leads to a loss in virulence (Fig 1Bii).

Encouraged by this result we repeated this experiment (Experiment 2). However, this time we saw very little difference in weight over time (until day 7 when mice start succumbing to infection) (Fig 1Ci). This repeat saw the exact opposite in time to death, whereby more mice survived in the vehicle treated cohort (i.e., no OTUD6B-AID knockdown), therefore our initial results are not readily reproducible (Fig 1Cii).

We believe the discrepancy between phenotypes observed in vitro following OTUD6B depletion and our mouse experiments is probably due to the caveats listed above, i.e. the RH strain is too virulent to see more subtle defects in a mouse model and/or that IAA does not lead to the same level of knockdown as occurs in tissue culture.

Overall, we feel like we cannot generate any firm conclusions from this set of experiments, especially with the caveats listed above.

We would also like to highlight that these experiments are very time consuming taking over a month per experiment and expensive (each mouse cost AUD\$22. i.e. $22 \times 48 = \$1056$) and weekly agistments costs of \$10 per cage per week ($10 \times 8 \text{ cages} \times 21 \text{ days} = \1680), which totalled \$2736 (not including peoples time). We also feel that any future mouse experiments are not ethically warranted.

We hope you understand the limitations of these experiments and therefore agree that they do not fit well in our manuscript.

Figure 1

Figure 1: Virulence testing of *Toxoplasma* AID mutants in C57B6 mice. Please see text for explanations.

2) OTUD6B-deficient *Toxoplasma* exhibited growth retardation in vitro (Figure 6). OTUD6B displayed specificity on K11 and K63 Ub linkages (Figure 3F). What *Toxoplasma* proteins are affected by the lack of OTUD6B? The authors should explore the substrate of OTUD6B by comparing wild-type and OTUD6B-deficient parasites using mass spectrometry. Among the list of potential substrates, the authors should discuss as to which protein(s) regulated by OTUD6B are important for the in vitro phenotype.

It's unclear what we would be looking for here? We have no reason to suspect what the substrates of OTUD6B are or that they have a defined biological function that would make the data meaningful. Furthermore, any changes that we found we would want to validate using orthogonal approaches, and furthermore, dissect the functional relevance of the identified substrates. This would take a huge amount of time.

(minor points)

3) Where is Figure 2D?

Have updated the text to state Table 2

4) Table S1 was not reader friendly. The table should include information of CRISPR fitness score and HyperLOPIT, which make reader understood at a glance.

This information is in Table 1 which is provided in Excel document "Tables.xlsx." Table S1 has been reformatted.

Reviewer #3 (Comments to the Authors (Required)):

The manuscript describes a comprehensive approach to identifying and characterizing OTU deubiquitinases of Toxoplasma. Several genes expression proteins with DUB activity were identified and characterized, including Ub linkage specificity, activity against UBLs, and functional dissection of accessory domains on some of the DUBs. The work provides an excellent foundation for further studies on OTU DUBs of Toxoplasma.

They first identified and listed the possible proteins in OTU DUB family of Toxoplasma, then predicted their 3-dimensional structures by using AlphaFold2. Based on this prediction, they compared these with OTU DUBs in other organisms, and designed constructs for in vitro studies. This paper makes excellent use of AlphaFold as a powerful data mining tool.

They identified whether catalytic sites of the cloned DUBs are active by using the probe Ub-HA with their proteins in vitro. Most of them showed ubiquitin binding, and interestingly showed pro-NEDD8 cleavage. In addition, they identified the DUBs' lysine linkage specificity of diUb. Some of them showed K6, K11, K48 and K63 specific cleavage, while TgOTU10 didn't show any of diUb cleavage even though it binds to Ub.

TgOTUD2, which has Zinc finger domain and UBX domain, resembles human OTU2. They attempted to identify the roles of these accessory domain, however only UBX domain seems responsible for substrate recognition of K63-linked diUb. Contrary to human OTU2, the ZF of TgOTU2 doesn't seem to affect substrate recognition.

TgOTU9 has a UBA-like accessory domain, which structure is similar to UBA domain of Dsk2 (which gives K48 specificity) in yeast, and the deletion of this domain abolished the K48 specific deubiquitinase activity of TgOTU9.

Finally, they attempted to knock down these DUBs (especially for TgOTU3B and TgOTU6B, based on the CRISPR screen score of previous report) in an AID model to identify their role in vivo. Suppression of TgOUT3B did not show effects on plaque growth, however TgOTU6B deletion led decrease in plaque size.

Overall, the paper is nicely written and provides important new contributions to parasitology Ub-biology.

General Comments

The manuscript is well written with lots of background and relevant topics related to the OTU domain deubiquitinases of Toxoplasma. The Figures are really nice.

1. There are several typos and mislabeled units however (Line 182, 493, Figure 5B and legends of Figure 6 and possibly more).

Fixed these and several others.

2. Table 1 is missing from in the manuscript, which is expected to show the CRISPR scores. This is one of the most important parts of this research, because they should show how they selected their candidates.

Tables 1 and 2 were uploaded as an Excel file which may not have been supplied to reviewers.

3. For Figure EV1 and manuscript, need to state how RMSDs were calculated (which atoms and regions were used for calculation). RMSD values appear quite small, which could be due to how much of each model was used in the calculations.

RMSDs were calculated using the Matchmaker tool in ChimeraX utilizing Needleman-Wunsch sequence alignment and iteration by pruning long atom pairs. The text in Figure legends for Figures 4, 5 and EV1 as well as methods section has been updated to describe this and indicate numbers of pruned atom pairs used for RMSD calculation. An additional Supplementary Table (S2) has been included which details RMSD values and numbers of atom pairs in each figure panel.

4. Knockdown of TgOTU3B in vivo results showed different result than expected, even though it was based on based on the previous screening results. Regarding this, CRISPR score value should be mentioned together (maybe it's on Table 1??). Suggest showing value of in vivo experiment data (means and errors etc) in the manuscript or figure to more clearly demonstrate differences.

This information is in Table 1, and text has been updated to include these values (Line 487).

5. Most of the tables are not completed and not properly titled. This needs to be addressed.

Tables have been completed and updated, Supplementary Table S1 has been reformatted to be more reader friendly.

6. Why was the linkage specificity limited to di-ubiquitin (diUb) substrates? Perhaps some of the DUBs cannot recognize di-Ub and thus some activities have been missed? This could

also explain lack of activity toward Ub-propargylamine by some of the enzymes as well perhaps?

In this study we tested a number of propargylamine probes (Ub, SUMO, NEDD8) and focused on diUb substrates as our lab has access to a full panel of diUb linkage substrates. Experiments performed with tri- and tetra-ubiquitin substrates are limited as we only have access to K6, K11, K48 and K63 linkages.

7. Minor points:

Line 40: 'and' NEDD8-based substrates? Replace 'but' with 'and'??

Changed.

Line 48; *Toxoplasma* (should be in italics)

Changed.

Line 71; The abbreviation for NFκB is missing

Text updated to include abbreviation. (Line 72)

Line 104 and 108; Abbreviations for some terms are missing

Text updated to include abbreviations. (Line 109)

Line 151 and 152; line spacing?

Fixed.

Line 158: Which software you used to generate the phylogenetic tree? (Figure 1A)

Phylogenetic tree was derived from a previously published study referenced in figure legend for Figure 1A, text has been updated to clarify this (Line 1267).

Line 165; Which software or method you used to interpret the data? Data source?

Domain identification and determination of domain boundaries/insertions was carried out using searches for PFAM IDs on VEuPathDB and searching sequences with PROSITE, and apicomplexan OTU DUB sequences were aligned against known related OTU domain sequences (identified through BLAST).

This has been addressed in Figure legend 1B (Line 1269) and Methods section (Line 635). MSAs for each OTU DUB are also available in Supplementary data 1.

Line 179, 185; Table 1 is not included in the supplementary document?

Tables 1 and 2 were provided in an Excel file which perhaps was not included in files sent to reviewers?

Line 179-181; Line spacing?

Fixed.

Line 215, 548; Appendix data S1 is not included in the section.

Appendix (renamed Supplementary Tables S1-S7) were included in the Appendix_PDF document

Line 276; How many constructs of full length and catalytic variants?

Text updated to reflect this (Line 315)

Line 391; Abbreviation for SCF?

Text updated to include abbreviation. (Line 453)

Line 458; Abbreviation for TNRF?

Text updated to include abbreviation. (Line 528)

Line 487; define ISG.

Text updated to include abbreviation. (Line 564)

Line 493; Typo error. Will need to be determined

Fixed (Line 570)

Line 534; Line spacing?

Fixed

Line 596; Table S2 is not included (primers?)

Line 637; Appendix Table S4 is not included

Line 645; Appendix Table S5 is not included

Line 667; Appendix Table S7 is not included

Line 689; Appendix Table S6 is not included

Appendix (renamed supplementary) tables S1-S8 are included in the Supplementary Tables excel file.

March 14, 2023

RE: Life Science Alliance Manuscript #LSA-2022-01710-TR

Prof. David Komander
Walter and Eliza Hall Institute of Medical Research
Ubiquitin signalling
1G Royal Parade
Parkville
Melbourne, Victoria 3052
Australia

Dear Dr. Komander,

Thank you for submitting your revised manuscript entitled "Characterisation of the OTU domain deubiquitinase complement of *Toxoplasma gondii*". We would be happy to publish your paper in Life Science Alliance pending final revisions necessary to meet our formatting guidelines.

- please add ORCID ID for secondary corresponding author-they should have received instructions on how to do so
- please make sure that the author order in the manuscript and the author order in our system match
- please add a figure callout for Figure 3A-B and Figure 5E to your main manuscript text
- please double-check your figure callouts for Figure S1; you have a callout for the panel K, but this isn't in the figure or the legend
- please incorporate your supp. Methods file into the main manuscript Materials and Methods section

A. FINAL FILES:

B. MANUSCRIPT ORGANIZATION AND FORMATTING:

Sincerely,

Reviewer #1 (Comments to the Authors (Required)):

The authors failed to test in vivo roles of OTUD3 and OTUD6 in the revised manuscript, however, this reviewer agrees with their reasons why they failed by looking at the supporting figures in the response-to-response letter. I warmly recommend its publication in LSA.

Reviewer #3 (Comments to the Authors (Required)):

The authors have addressed all of my concerns and I feel the manuscript contributes to the field and is ready for publication

March 15, 2023

RE: Life Science Alliance Manuscript #LSA-2022-01710-TRR

Prof. David Komander
Walter and Eliza Hall Institute of Medical Research
Ubiquitin signalling
1G Royal Parade
Parkville
Melbourne, Victoria 3052
Australia

Dear Dr. Komander,

Thank you for submitting your Research Article entitled "Characterisation of the OTU domain deubiquitinase complement of *Toxoplasma gondii*". It is a pleasure to let you know that your manuscript is now accepted for publication in Life Science Alliance. Congratulations on this interesting work.

DISTRIBUTION OF MATERIALS:

Again, congratulations on a very nice paper. I hope you found the review process to be constructive and are pleased with how the manuscript was handled editorially. We look forward to future exciting submissions from your lab.

Sincerely,
